# Uni-Mol: A Universal 3D Molecular Representation Learning Framework

## Abstract

Molecular representation learning (MRL) has gained tremendous attention due to its critical role in learning from limited supervised data for applications like drug design. In most MRL methods, molecules are treated as 1D sequential tokens or 2D topology graphs, limiting their ability to incorporate 3D information for downstream tasks and, in particular, making it almost impossible for 3D geometry prediction or generation. Herein, we propose Uni-Mol, a universal MRL framework that significantly enlarges the representation ability and application scope of MRL schemes. Uni-Mol is composed of two models with the same SE(3)-equivariant transformer architecture: a molecular pretraining model trained by 209M molecular conformations; a pocket pretraining model trained by 3M candidate protein pocket data. The two models are used independently for separate tasks, and are combined when used in protein-ligand binding tasks. By properly incorporating 3D information, Uni-Mol outperforms SOTA in 14/15 molecular property prediction tasks. Moreover, Uni-Mol achieves superior performance in 3D spatial tasks, including protein-ligand binding pose prediction, molecular conformation generation, etc. Finally, we show that Uni-Mol can be successfully applied to the tasks with few-shot data like pocket druggability prediction. The model and data will be made publicly available at `https://github.com/dptech-corp/Uni-Mol`.

## 1 Introduction

Recently, representation learning (or pretraining, self-supervised learning) [1, 2, 3] has been prevailing in many applications, such as BERT [4] and GPT [5, 6, 7] in Natural Language Processing (NLP), ViT [8] in Computer Vision (CV), etc. These applications have a common characteristic: unlabeled data is abundant, while labeled data is limited. As a solution, in a typical representation learning method, one first adopts a pretraining procedure to learn a good representation from large-scale unlabeled data, and then a finetuning scheme is followed to extract more information from limited supervised data.

Applications in the field of drug design share the characteristic that calls for representation learning schemes. The chemical space that a drug candidate lies in is vast, while drug-related labeled data is limited. Not surprisingly, compared with traditional molecular fingerprint based models [9, 10], recent molecular representation learning (MRL) models perform much better in most property prediction tasks [11, 12, 13]. However, to further improve the performance and extend the application scope of existing MRL models, one is faced with a critical issue. From the perspective of life science, the properties of molecules and the effects of drugs are mostly determined by their 3D structures [14, 15]. In most current MRL methods, one starts with representing molecules as 1D sequential strings, such as SMILES [16, 17, 18] and InChI [19, 20, 21], or 2D graphs [22, 11, 23, 12, 24]. This may limit their ability to incorporate 3D information for downstream tasks. In particular, this makes it almost impossible for 3D geometry prediction or generation, such as, e.g., the prediction of protein-

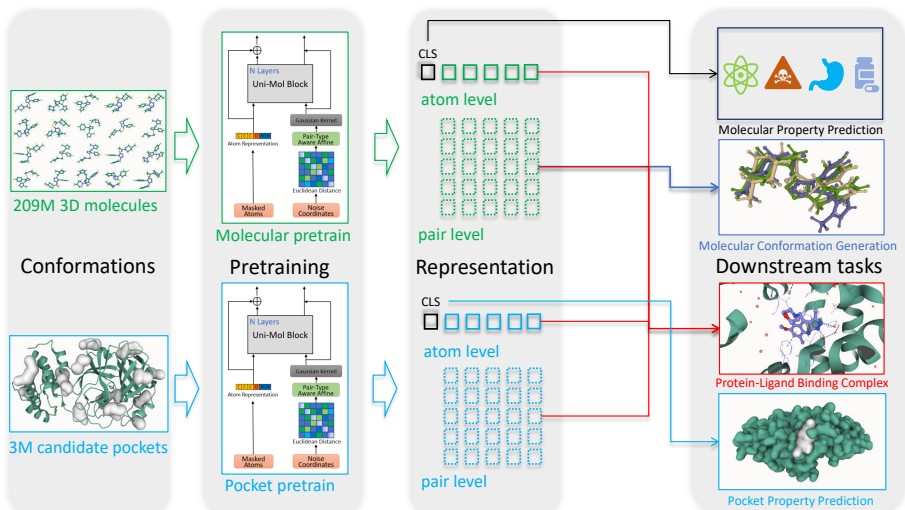

Figure 1: Schematic illustration of the Uni-Mol framework. Uni-Mol is composed of two models: a molecular pretraining model trained by 209M molecular 3D conformations; a pocket pretraining model trained by 3M candidate protein pocket data. The two models are used independently for separate tasks, and are combined when used in protein-ligand binding tasks.

ligand binding pose [25]. Even though there have been some recent attempts trying to leverage 3D information in MRL [26, 27], the performance is less than optimal, possibly due to the small size of 3D datasets, and 3D positions can not be used as inputs/outputs during finetuning, since they only serve as auxiliary information.

In this work, we propose Uni-Mol, to our best knowledge, the first universal 3D molecular pretraining framework, which is derived from large-scale unlabeled data and is able to directly take 3D positions as both inputs and outputs. Uni-Mol consists of 3 parts. 1) *Backbone*. Based on Transformer, the invariant spatial positional encoding and pair level representation are added to better capture the 3D information. Moreover, an equivariant head is used to directly predict 3D positions. 2) *Pretraining*. We create two large-scale datasets, a 209M molecular conformation dataset and a 3M candidate protein pocket dataset, for pretraining 2 models on molecules and protein pockets, respectively. For the pretraining tasks, besides masked atom prediction, a 3D position denoising task is used for learning 3D spatial representation. 3) *Finetuning*. According to specific downstream tasks, the used pretraining models are different. For example, in molecular property prediction tasks, only the molecular pretraining model is used; in protein-ligand binding pose prediction, both two pretraining models are used. We refer to Fig. 1 for an overall schematic illustration of the Uni-Mol framework.

To demonstrate the effectiveness of Uni-Mol, we conduct experiments on a series of downstream tasks. In the molecular property prediction tasks, Uni-Mol outperforms SOTA on 14/15 datasets on the MoleculeNet benchmark. In 3D geometric tasks, Uni-Mol also achieves superior performance. For the pose prediction of protein-ligand complexes, Uni-Mol predicts 88.07% binding poses with RMSD <= 2Å, 22.81% more than popular docking methods, and ranks 1st in the docking power test on CASF-2016 [28] benchmark. Regarding molecular conformation generation, Uni-Mol achieves SOTA for both Coverage and Matching metrics on GEOM-QM9 and GEOM-Drugs [29]. Moreover, Uni-Mol can be successfully applied to tasks with very limited data like pocket druggability prediction.

## 2   Uni-Mol Framework

In this section, we introduce the Uni-Mol framework by showing the details of the backbone, the pretraining scheme, and the finetuning scheme. We refer to Fig. 2 for a schematic illustration of the model architecture.

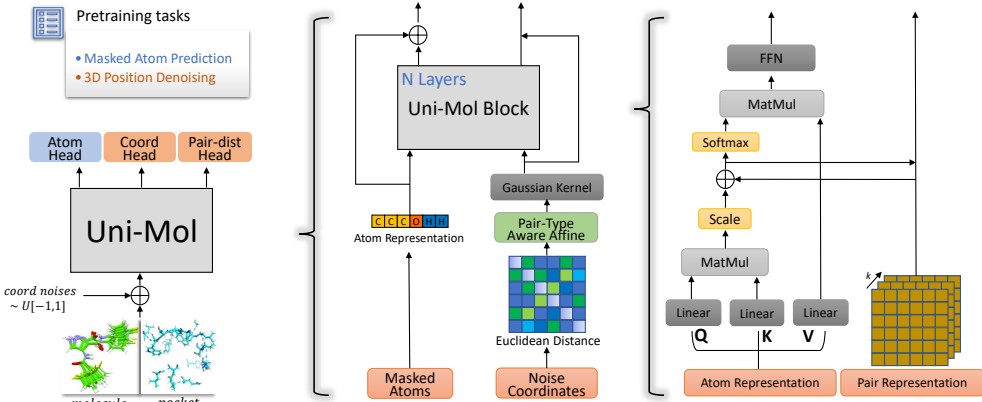

Figure 2: Left: the overall pretraining architecture. Middle: the model inputs, including atoms and spatial positional encoding created by pair Euclidean distance. Right: pair representation and its update process.

## 2.1 Backbone

Transformer [30] is widely used as a backbone model in representation learning. However, Transformer was originally designed for NLP tasks and cannot handle 3D spatial data directly. To tackle this, based on the standard Transformer with Pre-LayerNorm [31] backbone, we introduce several modifications.

**Invariant spatial positional encoding**  Due to its permutationally invariant property, Transformer cannot distinguish the positions of inputs without positional encoding. Different with the discrete (ordinal) positions used in NLP/CV [32, 33], the positions in 3D space, i.e. coordinates, are continuous values. Besides, the positional encoding procedure needs to be invariant under global rotation and translation. To achieve that, similar to the relative positional encoding, we simply use Euclidean distances of all atom pairs, as well as pair-type aware Gaussian kernels [34]. Formally, the $D$-channel positional encoding of atom pair $ij$ is denoted as

$$\boldsymbol{p}_{ij} = \{\mathcal{G}(\mathcal{A}(d_{ij}, t_{ij}; \boldsymbol{a}, \boldsymbol{b}), \mu^k, \sigma^k) | k \in [1, D]\}, \quad \mathcal{A}(d, r; \boldsymbol{a}, \boldsymbol{b}) = a_r d + b_r, \tag{1}$$

where $\mathcal{G}(d, \mu, \sigma) = \frac{1}{\sigma\sqrt{2\pi}} e^{-\frac{(d-\mu)^2}{2\sigma^2}}$ is a Gaussian density function with parameters $\mu$ and $\sigma$, $d_{ij}$ is the Euclidean distance of atom pair $ij$, and $t_{ij}$ is the pair-type of atom pair $ij$. Please note the pair-type here is not the chemical bond, and it is determined by the atom types of pair $ij$. $\mathcal{A}(d_{ij}, t_{ij}; \boldsymbol{a}, \boldsymbol{b})$ is the affine transformation with parameters $\boldsymbol{a}$ and $\boldsymbol{b}$, it affines $d_{ij}$ corresponding to its pair-type $t_{ij}$. Except $d_{ij}$ and $t_{ij}$, all remaining parameters are trainable and randomly initialized.

**Pair representation**  By default, Transformer maintains the token(atom) level representation, which is later used in finetuning downstream tasks. Nevertheless, as the spatial positions are encoded at pair-level, we also maintain the pair-level representation, to better learn the 3D spatial representation. Specifically, the pair representation is initialized as the aforementioned spatial positional encoding. Then, to update pair representation, we use the atom-to-pair communication via the multi-head Query-Key product results in self-attention. Formally, the update of $ij$ pair representation is denoted as

$$\boldsymbol{q}_{ij}^0 = \boldsymbol{p}_{ij} \boldsymbol{M}, \quad \boldsymbol{q}_{ij}^{l+1} = \boldsymbol{q}_{ij}^l + \{\frac{\boldsymbol{Q}_i^{l,h}(\boldsymbol{K}_j^{l,h})^T}{\sqrt{d}} | h \in [1, H]\}, \tag{2}$$

where $\boldsymbol{q}_{ij}^l$ is the pair representation of atom pair $ij$ in $l$-th layer, $H$ is the number of attention heads, $d$ is the dimension of hidden representations, $\boldsymbol{Q}_i^{l,h}$ ($\boldsymbol{K}_j^{l,h}$) is the Query (Key) of the $i$-th ($j$-th) atom in the $l$-th layer $h$-th head, and $\boldsymbol{M} \in \mathbb{R}^{D \times H}$ is the projection matrix to make the representation the same shape as multi-head Query-Key product results.

Besides, to leverage 3D information in the atom representation, we also introduce the pair-to-atom communication, by using the pair representation as the bias term in self-attention. Formally, the

self-attention with pair-to-atom communication is denoted as

$$\text{Attention}(\boldsymbol{Q}_i^{l,h}, \boldsymbol{K}_j^{l,h}, \boldsymbol{V}_j^{l,h}) = \text{softmax}(\frac{\boldsymbol{Q}_i^{l,h}(\boldsymbol{K}_j^{l,h})^T}{\sqrt{d}} + \boldsymbol{q}_{ij}^{l-1,h})\boldsymbol{V}_j^{l,h}, \tag{3}$$

where $\boldsymbol{V}_j^{l,h}$ is the Value of the $j$-th atom in the $l$-th layer $h$-th head. The pair representation and atom-pair communication are firstly proposed in the Evoformer in AlphaFold [35], but the cost of Evoformer is extremely large. In Uni-Mol, as we keep them as simple as possible, the extra cost of maintaining pair representation is negligible.

**SE(3)-Equivariance coordinate head**    With 3D spatial positional encoding and pair representation, the model can learn a good 3D representation. However, it still lacks the ability to directly output co-ordinates, which is essential in 3D spatial tasks. To this end, we add a simple SE(3)-equivariance head to Uni-Mol. Following the idea of EGNN [36], the design of SE(3)-equivariance head is denoted as

$$\hat{\boldsymbol{x}}_i = \boldsymbol{x}_i + \sum_{j=1}^{n} \frac{(\boldsymbol{x}_i - \boldsymbol{x}_j)c_{ij}}{n}, \quad c_{ij} = \text{ReLU}((\boldsymbol{q}_{ij}^L - \boldsymbol{q}_{ij}^0)\boldsymbol{U})\boldsymbol{W}, \tag{4}$$

where $n$ is the number of total atoms, $L$ is the number of layers in model, $\boldsymbol{x}_i \in \mathbb{R}^3$ is the input coordinate of $i$-th atom, and $\hat{\boldsymbol{x}}_i \in \mathbb{R}^3$ is the output coordinate of $i$-th atom, $\text{ReLU}(y) = \max(0, y)$ is Rectified Linear Unit [37], $\boldsymbol{U} \in \mathbb{R}^{H \times H}$ and $\boldsymbol{W} \in \mathbb{R}^{H \times 1}$ are the projection matrices to convert pair representation to scalar.

## 2.2    Pretraining

For the purpose of pretraining, we generate two large-scale datasets, one composed of 3D structures of organic molecules, and another composed of 3D structures of candidate protein pockets. Then, two models are pretrained using these two datasets, respectively. As pockets are directly involved in many drug design tasks, intuitively, the pretraining on candidate protein pockets can boost the performance of tasks related to protein-ligand structures and interactions.

The molecular pretraining dataset is based on multiple public datasets (See Appendix **??** for more information). After normalizing and deduplicating, it contains about 19M molecules. To generate 3D conformations, we use ETKGD [38] with Merck Molecular Force Field [39] optimization in RDKit [40] to randomly generate 10 conformations for each molecule. We also generate an additional 2D conformation (based on the molecular graph), to avoid some rare cases that fail to generate 3D conformations.

The protein pocket pretraining dataset is derived from the Protein Data Bank (RCSB PDB [1]) [41], a collection of 180K 3D structures of proteins. To extract candidate pockets, we first clean the data by adding the missing side chains and hydrogen atoms; then we use Fpocket [42] to detect possible binding pockets of the proteins; and finally, we filter pockets by the number of residues in contact with and retains water molecules in the pocket. In this way, We collect a dataset composed of 3.2M candidate pockets for pretraining.

Self-supervised task is vitally important for effective learning from large-scale unlabeled data. For example, the masked token prediction task in BERT [4] encourages the model to learn the contextual information. Similar to BERT, the masked atom prediction task is used in Uni-Mol. For each molecule/pocket, we add a special atom [CLS], whose coordinate is the center of all atoms, to represent the whole molecule/pocket. However, as 3D spatial positional encoding leaks chemical bonds, atom types could be inferred easily, and therefore, the masked atom prediction cannot encourage the model to learn useful information. To tackle this, as well as learning from 3D information, we design a 3D position denoising task. Particularly, uniform noises of [-1 Å, 1 Å] are added to the random 15% atom coordinates, then the spatial positional encoding is calculated based on corrupted coordinates. In this way, the masked atom prediction task becomes non-trivial. Besides, two additional heads are used to recover the correct spatial positions. 1) Pair-distance prediction. Based on pair-representation, the model needs to predict the correct Euclidean distances of the atoms pairs with corrupted coordinates. 2) Coordinate prediction. Based on SE(3)-Equivariance coordinate head, the model needs to predict the correct coordinates for the atoms with corrupted coordinates.

---

[1]http://www.rcsb.org/

141 Both 2 pretraining models use the same self-supervised tasks described above, and Figure 2 is the
142 illustration of the overall pretraining framework. For the detailed configurations of pretraining, please
143 refer to Appendix **??**.

## 2.3 Finetuning

145 To be consistent with pretraining, we use the same data prepossessing pipeline during finetuning.
146 For molecules, as multiple random conformations can be generated in a short time, we can use them
147 as data augmentation in finetuning to improve performance and robustness. Some molecules may fail
148 to generate 3D conformations, and we use their molecular graph as 2D conformation. For tasks that
149 provide atom coordinates, we use them directly and skip the 3D conformation generation process.
150 As there are 2 pretraining models and several types of downstream tasks, we should properly use
151 them in the finetuning stage. According to the task types, and the involvement of protein or ligand,
152 we can categorize them as follow.

**Non-3D prediction tasks**   These tasks do not need to output 3D conformations. Examples include
154 molecular property prediction, molecule similarity, pocket druggability prediction, protein-ligand
155 binding affinity prediction, etc. Similar to NLP/CV, we can simply use the representation of `[CLS]`
156 which represents the whole molecule/pocket, or the mean representation of all atoms, with a linear
157 head to finetune on downstream tasks. In the tasks with pocket-molecule pair, we can concatenate
158 their `[CLS]` representations, and then finetune with linear head.

**3D prediction tasks of molecules or pockets**   These tasks need to predict a 3D conformation
160 of the input, such as molecular conformation generation. Different with the fast conformation
161 generation method used in Uni-Mol, molecular conformation generation task usually requires running
162 advanced sampling and semi-empirical density functional theory (DFT) to account for the ensemble
163 of 3D conformers that are accessible to a molecule, and this is very time-consuming. Therefore,
164 there are many recent works that train the model to fast generate conformations from molecular
165 graph [43, 44, 45, 46]. While in Uni-Mol, this task straightforwardly becomes a conformation
166 optimization task: generate a new conformation based on a different input conformation. Specifically,
167 in finetuning, the model supervised learns the mapping from Uni-Mol generated conformations to
168 the labeled conformations. Moreover, the optimized conformations can be generated end-to-end by
169 SE(3)-Equivariance coordinate head.

**3D prediction tasks of protein-ligand pairs**   This is one of the most important tasks in structure-
171 based drug design. The task is to predict the complex structure of a protein binding site and a
172 molecular ligand. Besides the conformation changes of the pocket and the molecule themselves, we
173 also need to consider how the molecule lays in the pocket, that is, the additional 6 degrees (3 rotations
174 and 3 translations) of freedom of a rigid movement. In principle, with Uni-Mol, we can predict the
175 complex conformation by the SE(3)-Equivariant coordinate head in an end-to-end fashion. However,
176 this is unstable as it is very sensitive to the initial docking positions of molecular ligand. Herein, to
177 get rid of the initial positions, we use a scoring function based optimization method in this paper. In
178 particular, the molecular representation and pocket representation are firstly obtained from their own
179 pretraining models by their own conformations; then, their representations are concatenated as the
180 input of an additional 4-layer Uni-Mol decoder, which is finetuned to learn the pair distances of all
181 atoms in molecule and pocket. With the predicted pair-distance matrix as the scoring function, we
182 use a simple differential evolution algorithm [47] to sample and optimize the complex conformations.
183 More details can be found in Appendix **??**.

## 3 Experiments

185 To verify the effectiveness of our proposed Uni-Mol model, we conduct extensive experiments
186 on multiple downstream tasks, including molecular property prediction, molecular conformation
187 generation, pocket property prediction, and protein-ligand binding pose prediction. Besides, we also
188 conduct several ablation studies. Due to space restrictions, we leave the detailed experimental settings
189 and ablation studies to Appendix **??**.

### 3.1 Molecular property prediction

**Datasets and setup**   MoleculeNet [48] is a widely used benchmark for molecular property
192 prediction, including datasets focusing on different levels of properties of molecules, from quantum

Table 1: Uni-Mol performance on molecular property prediction classification tasks

| | Classification (ROC-AUC %, higher is better ↑) | | | | | | | | |
|---|---|---|---|---|---|---|---|---|---|
| Datasets | BBBP | BACE | ClinTox | Tox21 | ToxCast | SIDER | HIV | PCBA | MUV |
| # Molecules | 2039 | 1513 | 1478 | 7831 | 8575 | 1427 | 41127 | 437929 | 93087 |
| # Tasks | 1 | 1 | 2 | 12 | 617 | 27 | 1 | 128 | 17 |
| D-MPNN | 71.0(0.3) | 80.9(0.6) | 90.6(0.6) | 75.9(0.7) | 65.5(0.3) | 57.0(0.7) | 77.1(0.5) | 86.2(0.1) | 78.6(1.4) |
| Attentive FP | 64.3(1.8) | 78.4(0.022) | 84.7(0.3) | 76.1(0.5) | 63.7(0.2) | 60.6(3.2) | 75.7(1.4) | 80.1(1.4) | 76.6(1.5) |
| N-Gram$_{RF}$ | 69.7(0.6) | 77.9(1.5) | 77.5(4.0) | 74.3(0.4) | - | 66.8(0.7) | 77.2(0.1) | - | 76.9(0.7) |
| N-Gram$_{XGB}$ | 69.1(0.8) | 79.1(1.3) | 87.5(2.7) | 75.8(0.9) | - | 65.5(0.7) | 78.7(0.4) | - | 74.8(0.2) |
| PretrainGNN | 68.7(1.3) | 84.5(0.7) | 72.6(1.5) | 78.1(0.6) | 65.7(0.6) | 62.7(0.8) | 79.9(0.7) | 86.0(0.1) | 81.3(2.1) |
| GROVER$_{base}$ | 70.0(0.1) | 82.6(0.7) | 81.2(3.0) | 74.3(0.1) | 65.4(0.4) | 64.8(0.6) | 62.5(0.9) | 76.5(2.1) | 67.3(1.8) |
| GROVER$_{large}$ | 69.5(0.1) | 81.0(1.4) | 76.2(3.7) | 73.5(0.1) | 65.3(0.5) | 65.4(0.1) | 68.2(1.1) | 83.0(0.4) | 67.3(1.8) |
| GraphMVP | 72.4(1.6) | 81.2(0.9) | 79.1(2.8) | 75.9(0.5) | 63.1(0.4) | 63.9(1.2) | 77.0(1.2) | - | 77.7(0.6) |
| MolCLR | 72.2(2.1) | 82.4(0.9) | 91.2(3.5) | 75.0(0.2) | - | 58.9(1.4) | 78.1(0.5) | - | 79.6(1.9) |
| GEM | 72.4(0.4) | 85.6(1.1) | 90.1(1.3) | 78.1(0.1) | 69.2(0.4) | **67.2(0.4)** | 80.6(0.9) | 86.6(0.1) | 81.7(0.5) |
| Uni-Mol | **72.9(0.6)** | **85.7(0.2)** | **91.9(1.8)** | **79.6(0.5)** | **69.6(0.1)** | 65.9(1.3) | **80.8(0.3)** | **88.5(0.1)** | **82.1(1.3)** |

Table 2: Uni-Mol performance on molecular property prediction regression tasks

| | Regression (lower is better ↓) | | | | | |
|---|---|---|---|---|---|---|
| | RMSE | | | MAE | | |
| Datasets | ESOL | FreeSolv | Lipo | QM7 | QM8 | QM9 |
| # Molecules | 1128 | 642 | 4200 | 6830 | 21786 | 133885 |
| # Tasks | 1 | 1 | 1 | 1 | 12 | 3 |
| D-MPNN | 1.050(0.008) | 2.082(0.082) | 0.683(0.016) | 103.5(8.6) | 0.0190(0.0001) | 0.00814(0.00001) |
| Attentive FP | 0.877(0.029) | 2.073(0.183) | 0.721(0.001) | 72.0(2.7) | 0.0179(0.001) | 0.00812(0.00001) |
| N-Gram$_{RF}$ | 1.074(0.107) | 2.688(0.085) | 0.812(0.028) | 92.8(4.0) | 0.0236(0.0006) | 0.01037(0.00016) |
| N-Gram$_{XGB}$ | 1.083(0.082) | 5.061(0.744) | 2.072(0.030) | 81.9(1.9) | 0.0215(0.0005) | 0.00964(0.00031) |
| PretrainGNN | 1.100(0.006) | 2.764(0.002) | 0.739(0.003) | 113.2(0.6) | 0.0200(0.0001) | 0.00922(0.00004) |
| GROVER$_{base}$ | 0.983(0.090) | 2.176(0.052) | 0.817(0.008) | 94.5(3.8) | 0.0218(0.0004) | 0.00984(0.00055) |
| GROVER$_{large}$ | 0.895(0.017) | 2.272(0.051) | 0.823(0.010) | 92.0(0.9) | 0.0224(0.0003) | 0.00986(0.00025) |
| GraphMVP | 1.029(0.033) | - | 0.681(0.010) | - | - | - |
| MolCLR | 1.271(0.040) | 2.594(0.249) | 0.691(0.004) | 66.8(2.3) | 0.0178(0.0003) | - |
| GEM | 0.798(0.029) | 1.877(0.094) | 0.660(0.008) | 58.9(0.8) | 0.0171(0.0001) | 0.00746(0.00001) |
| Uni-Mol | **0.788(0.029)** | **1.620(0.035)** | **0.603(0.010)** | **41.8(0.2)** | **0.0156(0.0001)** | **0.00467(0.00004)** |

mechanics and physical chemistry to biophysics and physiology. Following previous work GEM [13], we use scaffold splitting for the dataset and report the mean and standard deviation of the results for three random seeds.

**Baselines** We compare Uni-Mol with multiple baselines, including supervised and pretraining baselines. D-MPNN [49] and AttentiveFP [50] are supervised GNNs methods. N-gram [51], PretrainGNN [22], GROVER [11], GraphMVP [26], MolCLR [12], and GEM [13] are pretraining methods. N-gram embeds the nodes in the graph and assembles them in short walks as the graph representation. Random Forest and XGBoost [52] are used as the predictor for downstream tasks.

**Results** Table 1 and Table 2 show the experiment results of Uni-Mol and competitive baselines, where the best results are marked in bold. Most baseline results are from the paper of GEM, except for the recent works GraphMVP and MolCLR. The results of GraphMVP are from its paper. As MolCLR uses a different data split setting (without considering chirality), we rerun it with the same data split setting as other baselines. From the results, we can summarize them as follows: 1) overall, Uni-Mol outperforms baselines on almost all downstream datasets. 2) In solubility (ESOL, Lipo), free energy (FreeSolv), and quantum mechanical (QM7, QM8, QM9) properties prediction tasks, Uni-Mol is significantly better than baselines. As 3D information is critical in these properties, it indicates that Uni-Mol can learn a better 3D representation than other baselines. 3) Uni-Mol fails to beat SOTA on the SIDER dataset. After investigation, we find Uni-Mol fails to generate 3D conformations (and rollbacks to 2D graphs) for many molecules (like natural products and peptides) in SIDER. Therefore, due to the missing 3D information, it is reasonable that Uni-Mol cannot outperform others.

In summary, by better utilizing 3D information in pretraining, Uni-Mol outperforms all previous MRL models in almost all property prediction tasks.

Table 3: Uni-Mol performance on molecular conformation generation

| Dataset | QM9 | | | | Drugs | | | |
|---|---|---|---|---|---|---|---|---|
| | COV(↑, %) | | MAT(↓, Å) | | COV(↑, %) | | MAT(↓, Å) | |
| Methods | Mean | Median | Mean | Median | Mean | Median | Mean | Median |
| RDKit | 83.26 | 90.78 | 0.3447 | 0.2935 | 60.91 | 65.70 | 1.2026 | 1.1252 |
| CVGAE | 0.09 | 0.00 | 1.6713 | 1.6088 | 0.00 | 0.00 | 3.0702 | 2.9937 |
| GraphDG | 73.33 | 84.21 | 0.4245 | 0.3973 | 8.27 | 0.00 | 1.9722 | 1.9845 |
| CGCF | 78.05 | 82.48 | 0.4219 | 0.3900 | 53.96 | 57.06 | 1.2487 | 1.2247 |
| ConfVAE | 80.42 | 85.31 | 0.4066 | 0.3891 | 53.14 | 53.98 | 1.2392 | 1.2447 |
| ConfGF | 88.49 | 94.13 | 0.2673 | 0.2685 | 62.15 | 70.93 | 1.1629 | 1.1596 |
| GeoMol | 71.26 | 72.00 | 0.3731 | 0.3731 | 67.16 | 71.71 | 1.0875 | 1.0586 |
| DGSM | 91.49 | 95.92 | 0.2139 | 0.2137 | 78.73 | 94.39 | 1.0154 | 0.9980 |
| DMCG | 96.34 | 99.53 | 0.2065 | 0.2003 | **96.69** | 100.00 | 0.7223 | 0.7236 |
| GeoDiff | 90.07 | 93.39 | 0.2090 | 0.1988 | 89.13 | 97.88 | 0.8629 | 0.8529 |
| **Uni-Mol** | **98.68** | **100.00** | **0.1806** | **0.1510** | 92.69 | **100.00** | **0.6596** | **0.6215** |

## 3.2 Molecular conformation generation

**Datasets and setup** Following the settings in previous works [44, 53], we use GEOM-QM9 and GEOM-Drugs [54] dataset to perform conformation generation experiments. As described in Sec. 2.3, in this task, Uni-Mol optimizes its generative conformations to the labeled ones. To construct the finetuning data, we first randomly generate 10 conformations. Then, for each of them, we calculate the RMSD between it and labeled conformations, and choose the one with minimal RMSD as its optimizing target. For the inference in the test set, we generate the same number of conformations (twice the number of labeled conformations) as previous works do. And we use the same metrics, Coverage (COV) and Matching (MAT). Higher COV means better diversity, while lower MAT means higher accuracy.

**Baselines** We compare Uni-Mol with 10 competitive baselines. RDKit [38] is a traditional conformation generation method based on distance geometry. The rest baseline can be categorized into two classes. GraphDG [43], CGCF[44], ConfVAE [55], ConfGF [53], and DGSM [56] combine generative models with distance geometry, which first generates interatomic distance matrices and then iteratively generates atomic coordinates. CVGAE [45], GeoMol [46], DMCG [57], and GeoDiff [58] directly generate atomic coordinates.

**Results** The results are shown in Table 3. We report the mean and median of COV and MAT on GEOM-QM9 and GEOM-Drugs datasets. ConfVAE [55], GeoMol[46], DGSM [56], DMCG [57], GeoDiff's [58] results are from their papers, respectively. Other baseline results are from ConfGF's paper. As shown in Table 3, Uni-Mol exceeds existing baselines in both COV and MAT metrics on both datasets. Although Uni-Mol outperforms SOTA, we suspect that the above benchmark cannot satisfy the real-world demand of conformation generation tasks in the field of drug design. Since the ensemble of molecular conformations in biological systems is different from that in a vacuum or general solution environment, the ensemble of bioactive conformation must be considered in order to apply the conformation generation model in the context of drug design, while the GEOM dataset just ignores this. Establishing a reasonable benchmark will be crucial in this research direction.

## 3.3 Pocket property prediction

**Datasets and setup** Druggability, the ability of a candidate protein pocket to produce stable binding to a specific molecular ligand, is one of the most critical properties of a candidate protein pocket. However, this task is very challenging due to the very limited supervised data. For example, NRDLD [59], a commonly used dataset, only contains 113 data samples. Therefore, besides NRDLD, we construct a regression dataset for benchmarking pocket property prediction performance. Specifically, based on Fpocket tool, we calculate Fpocket Score, Druggability Score, Total SASA, and Hydrophobicity Score for the selected 164,586 candidate pockets. Model is trained to predict these scores. To avoid leaking, the selected pockets are not overlapped with the candidate protein pocket dataset used in Uni-Mol pretraining.

**Baselines** On the NRDLD dataset, we compare Uni-Mol with 6 previous methods evaluated in [60]. Accuracy, recall, precision, and F1-score are used as metrics for this classification task. On our created benchmark dataset, as there are no appropriate baselines, we use an additional Uni-Mol model

Table 4: Uni-Mol performance on pocket property prediction

| Dataset | Classification (higher is better ↑) NRDLD | | | | | | Regression (lower is better ↓) Fpocket Scores | | |
|---|---|---|---|---|---|---|---|---|---|
| Methods | Cavity-DrugScore | Volsite | DrugPred | PockDrug | TRAPP-CNN | **Uni-Mol** | Methods | Uni-Mol$_{random}$ | **Uni-Mol** |
| Accuracy | 0.82 | 0.89 | 0.89 | 0.865 | **0.946** | **0.946** | MSE$_{Fpocket}$ | 0.621(0.004) | **0.551(0.008)** |
| Recall | - | - | - | 0.957 | 0.913 | **1.000** | MSE$_{Druggability}$ | 0.601(0.02) | **0.499(0.007)** |
| Precision | - | - | - | 0.846 | **1.000** | 0.920 | MSE$_{Total\ SASA}$ | 0.197(0.008) | **0.129(0.005)** |
| F1-score | - | - | - | 0.898 | 0.955 | **0.958** | MSE$_{Hydrophobicity}$ | 0.0357(0.017) | **0.0127(0.0005)** |

without pretraining, denoted as Uni-Mol$_{random}$, to check the performance brought by pretraining on pocket property prediction. MSE (mean square error) is used as the metric.

**Results**  As shown in Table 4, Uni-Mol shows the best accuracy, recall, and F1-score on NRDLD, the few-show dataset. In our created benchmark dataset, the pretraining Uni-Mol model largely outperforms the non-pretraining one on all four scores. This indicates that pretraining on candidate protein pockets indeed brings improvement in pocket property prediction tasks.

Unlike Molecular property prediction, due to the very limited supervised data, pocket property prediction gained much less attention. Therefore, we also plan to release our created benchmark dataset, and hopefully, it can help future research.

### 3.4 Protein-ligand binding pose prediction

**Datasets and setup**  As mentioned above, protein-ligand binding pose prediction is one of the most important tasks in drug design. And Uni-Mol combines both the molecular and pocket pretraining models to learn a distance matrix based scoring function, and then sample and optimize the complex conformations. For the benchmark dataset, referring to the previous works [28, 61], we use CASF-2016 as the test set. For the training data used in finetuning, we use PDBbind General set v.2020 [62] (19,443 protein-ligand complexes), excluding complexes that already exist in the CASF-2016.

Two benchmarks are conducted: 1) Docking power, the default metric to benchmark the ability of a scoring function in CASF-2016. Specifically, it tests whether a scoring function can distinguish the ground truth binding pose from a set of decoys or not. For each ground truth, CASF-2016 provides 50 100 decoy conformations of the same ligand. Scoring functions are applied to rank them, and the ground truth binding pose is expected to be the top 1. 2) Binding pose accuracy. Specifically, we use the semi-flexible docking setting: keep the pocket conformation fixed, while the conformation of the ligand is fully flexible. We evaluate the RMSD between the predicted binding pose and the ground truth. Following previous works, we use the percentage of results that are below predefined RMSD thresholds as metrics.

**Baselines**  For docking power benchmark, the baselines are DeepDock [61] and the top 10 scoring functions reported in [28], including both conventional scoring functions and machine learning-based ones. For the binding pose accuracy, the baselines are Autodock Vina [63, 64], Vinardo [65], Smina [66], and AutoDock4 [67].

**Results**  From the docking power benchmark results shown in Figure 3, Uni-Mol ranks the 1st, with the top 1 success rate of 91.6%. For comparison, the previous top scoring function AutoDock Vina [63, 64] achieves 90.2% of the top 1 success rate in this benchmark. From the binding pose accuracy results shown in Table 5, Uni-Mol also surpasses all other baselines. Notably, Uni-Mol outperforms the second best method by 22.81% under the threshold of 2Å. This result indicates that Uni-Mol can effectively learn the 3D information from both molecules and pockets, as well as the interaction in 3D space of them. Even without pretraining, Uni-Mol (denoted as Uni-Mol$_{random}$) is also better than other baselines. This demonstrates the effectiveness of Uni-Mol backbone, as it effectively learns the 3D information by limited data.

In summary, by combining molecular and pocket pretraining models, Uni-Mol significantly outperforms the widely used docking tools in the protein-ligand binding tasks.

## 4  Related work

**Molecular representation learning**  Representation learning on large-scale unlabeled molecules attracts much attention recently. SMILES-BERT [18] is pretrained on SMILES strings of molecules using BERT [4]. Subsequent works are mostly pretraining on 2D molecular topological graphs [23, 11]. MolCLR [12] applies data augmentation to molecular graphs at both node and graph levels, using

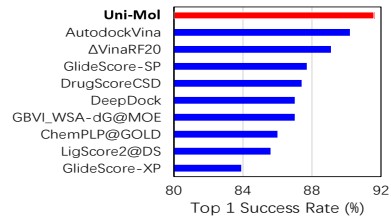

Figure 3: Docking power evaluation on CASF-2016 (Top 10 methods)

| | Ligand RMSD | | | | | |
| | % Below Threshold ↑ | | | | | |
| Methods | 0.5 Å | 1.0 Å | 1.5 Å | 2.0 Å | 3.0 Å | 5.0 Å |
| Autodock Vina | 23.86 | 44.21 | 57.54 | 64.56 | 73.68 | 84.56 |
| Vinardo | 23.51 | 41.75 | 57.54 | 62.81 | 69.82 | 76.84 |
| Smina | 23.51 | 47.37 | 59.65 | 65.26 | 74.39 | 82.11 |
| Autodock4 | 7.02 | 21.75 | 31.58 | 35.44 | 47.02 | 64.56 |
| Uni-Mol$_{random}$ | 14.04 | 49.47 | 65.26 | 75.44 | 87.02 | 98.60 |
| **Uni-Mol** | **24.91** | **70.53** | **84.21** | **88.07** | **94.74** | **98.95** |

Table 5: Uni-Mol performance on binding pose prediction

a self-supervised contrastive learning strategy to learn molecular representations. Further, several recent works try to leverage the 3D spatial information of molecules, and focus on contrastive or transfer learning between 2D topology and 3D geometry of molecules. For example, GraphMVP [26] proposes a contrastive learning GNN-based framework between 2D topology and 3D geometry. GEM [13] uses bond angles and bond length as additional edge attributes to enhance 3D information. As aforementioned, due to the inability of handling 3D information, most previous representation learning models cannot be used in the important 3D prediction tasks.

**SE(3)-Equivariant models**    In many-body scenarios such as potential energy surface fitting, SE-(3) equivariance is usually required. A series of SE(3) models are proposed, such as SchNet [68], tensor field networks [69], SE(3) Transformer [70], DimmNet [71], equivariant graph neural networks (EGNN) [36], GemNet [72] and SphereNet [73]. Most of these models are used in supervised learning with energy and force. In Uni-Mol, based on the standard Transformer, we introduce several minor changes to make the model SE(3)-Equivariant.

**Pocket druggability prediction**    Druggability prediction of protein binding pockets is crucial for drug discovery as druggable pockets need to be identified at the beginning. Since proteins undergo conformation changes that might alter the druggability of pockets, it is necessary to utilize 3D spatial data beyond sequential information. Early methods, such as Volsite [74], DrugPred [59], and PockDrug [75], predict druggability based on the predefined descriptors of pockets' static structures. Later, TRAPP-CNN [60], based on 3D-CNN, proposes the analysis of proteins' conformation changes and the use of such information for druggability prediction.

**Protein-ligand binding pose prediction**    In structure-based drug design, it is crucial to understand the interactions between protein targets and ligands. The *in vitro* estimation of the binding pose and affinity, such as docking, allows for lead identification and guides molecular optimization. In particular, docking is one of the most important approaches in structure-based drug design and has been developed for the past decades. Tools such as AutoDock4 [67], AutoDock Vina [63, 64], and Smina [66] are among the most used docking programs. Also, machine learning-based docking methods, such as $\Delta_{Vina}RF_{20}$ [76], DeepDock [61] and Equibind [77], have also been developed to predict protein-ligand binding poses and assess protein-ligand binding affinity.

## 5    Conclusion

In this paper, to enlarge the application scope and representation ability of molecular representation learning (MRL), we propose Uni-Mol, the first universal large-scale 3D MRL framework. Uni-Mol consists of 3 parts: a Transformer based backbone to handle 3D data; two large-scale pretraining models to learn molecular and pocket representations respectively; finetuning strategies for all kinds of downstream tasks. Experiments demonstrate that Uni-Mol can outperform existing SOTA in various downstream tasks, especially in 3D spatial tasks.

There are 3 potential future directions. 1) Better interaction mechanisms for finetuning two pretraining models together. As the interaction between the pretraining pocket model and the pretraining molecular model is simple in the current version of Uni-Mol, we believe there is a large room for further improvement. 2) Large Uni-Mol models. As larger pretraining models often perform better, it is worthy of training a large Uni-Mol model on a bigger dataset. 3) More high-quality benchmarks. Although there have been many applications in the field of drug design, high-quality public datasets have been lacking. Many public datasets cannot satisfy real-world demand due to the low data quality. We believe the high-quality benchmarks will be the lighthouse of the entire field, and will significantly accelerate the development of drug design.

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
