# Appendix of Submission 3131:
# Uni-Mol: A Universal 3D Molecular Representation Learning Framework

## 1 Pretraining data

**Molecular dataset**   The pretraining datasets we use consist of two parts: one part is a database collection of 12 million molecules that can be synthesized and purchased (See Table 1), and the other part is taken from a previous work [1], whose molecules are collected from the ZINC [2] and ChemBL [3] databases. After normalizing and duplicating, we obtain 19 million molecules as our pretraining dataset. For each molecule, we add random conformer augmentations with ten 3D conformers generated by RDKit and one 2D graph to avoid ETKDG patterns missing match.

**Candidate protein pocket dataset**   The pretraining dataset for candidate protein pockets is derived from the Protein Data Bank (RCSB PDB [1]) [4], a collection of 180K structural data of proteins. We first pre-process the raw data by adding missing side chains and hydrogen atoms, and then we use Fpocket [5] to detect candidate binding pockets of the proteins. After filtering the raw pockets by the number of residues they have contact with (10~25) and including water molecules inside the pockets, we collect a pretraining dataset of 3,291,739 candidate pockets.

## 2 Downstream data supplements

**Molecular property prediction**   We conduct experiments on the MoleculeNet[6] benchmark in the molecular property prediction task. MoleculeNet is a widely used benchmark for molecular property prediction. The details of the 15 datasets we used are described below.

- **BBBP** Blood-brain barrier penetration (BBBP) contains the ability of small molecules to penetrate the blood-brain barrier.

- **BACE** This dataset contains the results of small molecules as inhibitors of binding to human $\beta$-secretase 1 (BACE-1).

- **ClinTox** This dataset contains the toxicity of the drug in clinical trials and the status of the drug for FDA approval[7].

- **Tox21** The dataset contains toxicity measurements of 8k molecules for 12 targets.

- **ToxCast** This dataset is derived from toxicology data from in vitro high-throughput screening and contains toxicity measurements for 8k molecules against 617 targets.

- **SIDER** The Side Effect Resource (SIDER) contains side effects of drugs on 27 system organs. These drugs are not only small molecules but also some peptides with molecular weights over 1000.

- **HIV** This dataset contains 40k compounds with the ability to inhibit HIV replication.

---

[1]http://www.rcsb.org/

Submitted to 36th Conference on Neural Information Processing Systems (NeurIPS 2022). Do not distribute.

Table 1: Database collection of 12M purchasable molecules

| Database | Molecules | Link |
|---|---|---|
| Targetmol | 10,000 | https://www.targetmol.com/ |
| Chemdiv | 1,613,931 | https://www.chemdiv.com/ |
| Enamine | 2,734,581 | https://enamine.net/ |
| Chembridge | 1,557,942 | https://www.chembridge.com/ |
| Life Chemical | 509,975 | https://lifechemicals.com/ |
| Specs | 208,670 | https://www.specs.net/ |
| Vitas-M | 1,409,339 | https://vitasmlab.biz/ |
| InterBioScreen | 48,627 | https://www.ibscreen.com/ |
| Maybridge | 53,352 | https://www.thermofisher.in/ |
| Bionet-Key Organics | 259,244 | https://www.keyorganics.net/ |
| Asinex | 530,881 | https://www.asinex.com/ |
| UkrOrgSynthesis | 688,952 | https://uorsy.com/ |
| Eximed | 61,009 | https://eximedlab.com/ |
| HTS Biochemie Innovationen | 58,437 | https://www.hts-biochemie.de/ |
| Princeton BioMolecular | 1,532,542 | https://princetonbio.com/ |
| Otava | 270,835 | https://otavachemicals.com/ |
| Alinda Chemical | 202,332 | https://www.alinda.ru/ |
| Analyticon | 42,664 | https://www.analyticon-diagnostics.com/ |

- **PCBA** PubChem BioAssay (PCBA) is a database of small molecule bioactivities generated by high-throughput screening. This is a subset containing over 400k molecules on 128 bioassays.

- **MUV** Maximum Unbiased Validation (MUV) is another subset of PubChem BioAssay, containing 90k molecules and 17 bioassays.

- **ESOL** This dataset contains the water solubility of the compound and is a small dataset with 1128 molecules.

- **FreeSolv** The dataset contains hydration free energy data for small molecules, of which we use the experimental values as labels.

- **Lipo** Lipophilicity contains the solubility of small molecules in lipids, of which we use the octanol/water distribution coefficient as the label.

- **QM7, QM8, QM9** The molecule in QM7 contains up to 7 heavy atoms, QM8 is 8 and QM9 is 9. These datasets provide the geometric, energetic, electronic and thermodynamic properties of the molecule, which are calculated by density functional theory (DFT)[8]. QM9 contains several quantum mechanical properties of different quantitative ranges, and we select *homo*, *lumo* and *gap* of similar quantitative range, following the setup of the previous work[9].

**Molecular corformation generation**   Following the settings in previous works [10, 11], we use GEOM-QM9 and GEOM-Drugs [12] dataset in this task.

- **GEOM** This dataset contains 37 million accurate conformations generated for 450,000 molecules by advanced sampling and semi-empirical density flooding theory (DFT). Of these, 133,000 molecules are from QM9, and the remaining 317,000 molecules have biophysical, physiological, or physical chemistry experimental data, i.e., Drugs.

**Pocket property prediction**   NRDLD [13] is a benchmark dataset for pocket druggability prediction. As NRDLD and other existing benchmark datasets are too small, we construct a regression dataset to benchmark pocket property prediction performance.

- **NRDLD** NRDLD contains 113 proteins, and a predefined split is provided: 76 proteins constitute the training set and 37 proteins constitute the test set. It labels 71 proteins as druggable in that they noncovalently bind small drug-like ligands [14]. The rest 42 proteins are labeled as less-druggable because none of the ligands they cocrystallized satisfy the following requirements simultaneously: the rule of five, clogP $\geq$ -2, and ligand efficiency, as defined in  [15], $\geq$ 0.3 kcal mol$^{-1}$ / heavy atom.

- **Our created benchmark dataset** The dataset contains 164,586 candidate pockets, and Fpocket scores each one of them on Fpocket Score, Druggability Score, Total SASA, and Hydrophobicity

Table 2: Uni-Mol hyperparameters setup during pre-training

| Hyperparameter | Molecular pretraining | Pocket pretraining |
|---|---|---|
| Layers | 15 | 15 |
| Peak learning rate | 1e-4 | 1e-4 |
| Batch size | 128 | 128 |
| Max training steps | 1M | 1M |
| Warmup steps | 10K | 10k |
| Attention heads | 64 | 64 |
| FFN dropout | 0.1 | 0.1 |
| Attention dropout | 0.1 | 0.1 |
| Embedding dropout | 0.1 | 0.1 |
| Weight decay | 1e-4 | 1e-4 |
| Embedding dim | 512 | 512 |
| FFN hidden dim | 2048 | 2048 |
| Gaussian kernel channels | 128 | 128 |
| Mask ratio | 0.15 | 0.15 |
| Coordinate noise | Uniform [-1 Å, 1 Å] | Uniform [-1 Å, 1 Å] |
| Activation function | GELU | GELU |
| Learning rate decay | Linear | Linear |
| Adams $\epsilon$ | 1e-6 | 1e-6 |
| Adams $(\beta_1, \beta_2)$ | (0.9, 0.99) | (0.9, 0.99) |
| Gradient clip norm | 1.0 | 1.0 |
| Atom loss function and its weight | Cross entropy, 1.0 | Cross entropy, 1.0 |
| Coordinate loss function and its weight | Smooth L1, 5.0 | Smooth L1, 1.0 |
| Distance loss function and its weight | Smooth L1, 10.0 | Smooth L1, 1.0 |
| Max number of atoms | 256 | 256 |
| Vocabulary size (atom types) | 30 | 9 |

Score. These four scores are indicators of the druggability of candidate pockets. To avoid leaking, the selected pockets are not overlapped with the candidate protein pocket dataset used in Uni-Mol pretraining.

**Protein-ligand binding pose prediction**  We use PDBbind General set v.2020 [16], excluding the complexes in CASF-2016 [17], as the training set. And CASF-2016 is used as the test set. In particular, we define the pocket for each protein-ligand pair as residues of the protein which have at least one atom within the range of 6Å from a heavy atom in the ligand. All atoms of the selected residues are included. In addition, we draw the smallest bounding box covering all of the atoms in the pocket and regard the water molecules in the bounding box as a part of the pockets, too.

• **PDBbind General set v.2020** This dataset contains 19,443 protein-ligand complexes with binding data and processed structural files originally from the Protein Data Bank (PDB). Only complexes with experimentally determined binding affinity data are included in the general set.

• **CASF-2016** CASF-2016 is the widely used benchmark for docking and scoring. This dataset, whose primary test set is known as the PDBbind Core set, contains 285 protein-ligand complexes with high quality crystal structures and reliable binding constants from PDBbind General set. For each protein-ligand complex, CASF-2016 provides 50~100 decoy molecular conformations of the same ligand for evaluation.

# 3 Experiments details & reproduce

**Molecular Pretraining setup**  We report the detailed hyperparameters setup of Uni-mol during pretraining in Table 2. Uni-Mol training loss is summed up by three components, atom(token) loss, coordinate loss, and pair-distance loss. Atoms are masked, and noise is added to coordinate as described in sections 2.1 and 2.2. Since the values of the above three components differ significantly, to make them have a similar influence, we enlarge the coordinate loss and distance loss.

**Pocket Pretraining setup**  The pocket Uni-Mol model is slightly different from molecule ones during pretraining: 1) We use a residue-level masking strategy instead of the original atom-level, as

Table 3: Search space for small datasets: BBBP, BACE, ClinTox, Tox21, Toxcast, SIDER, ESOL, FreeSolv, Lipo, QM7, QM8, for large datasets: PCBA, MUV, QM9, and for HIV

| Hyperparameter | Small | Large | HIV |
|---|---|---|---|
| Learning rate | [5e-5, 1e-4, 4e-4, 5e-4] | [2e-5, 1e-4] | [2e-5, 5e-5] |
| Batch size | [32, 64, 128, 256] | [128, 256] | [128, 256] |
| Epochs | [40 ,60, 80, 100] | [20, 40] | [2, 5, 10] |
| Pooler dropout | [0.0, 0.1, 0.2, 0.5] | [0.0, 0.1] | [0.0, 0.2] |
| Warmup ratio | [0.0, 0.06, 0.1] | [0.0, 0.06] | [0.0, 0.1] |

Table 4: Hyperparameters setup for molecular conformation generation

| | |
|---|---|
| Learning rate | 1e-4 |
| Batch size | 8 |
| Epochs | 5 |
| Warmup ratio | 0.06 |
| Coordinate loss function and weight | MSE, 1.0 |
| Distance loss function and weight | MSE, 1.0 |

residue granularity is non-redundancy and integrity in protein. 2) Only polar hydrogen is remained in pocket Uni-Mol pretraining, to reduce the number of used atoms and thus improve efficiency. 3) All weights of loss functions are set 1, as the residue-level masking strategy makes the 3D denoising task much harder. Other settings are listed in Table 2.

**Molecular property prediction**

- **Data split** In our experiments, referring to previous work GEM[9], we use scaffold splitting[18] to divide the dataset into training, validation, and test sets in the ratio of 8:1:1. Scaffold splitting is more challenging than random splitting as the scaffold sets of molecules in different subsets do not intersect. This splitting tests the model's generalization ability and reflects the realistic cases[6]. Since this splitting is according to the scaffold of the molecule, we find that whether or not chirality is considered when generating the scaffold using RDKit has a significant impact on the division results. From the results, the splitting considering chirality makes the task harder. The original implementation of MolCLR does not consider chirality, and we reproduce the experiment by considering it. In all experiments, we choose the checkpoint with the best validation loss, and report the results on the test-set run by that checkpoint.

- **Hyperparameter search space** Referring to previous works, we use a grid search to find the best combination of hyperparameters for the molecular property prediction task. To reduce the time cost, we set a smaller search space for the large datasets. The specific search space is shown in Table 3.

**Molecular conformation generation** We report the detailed hyperparameters setup for molecular conformation generation in Table 4. Since this is a 3D-related task, we only use coordinate loss and distance loss.

- **Data details** We leverage RDKit (ETKGD) for generating inputs in molecular conformation generation tasks. Specifically, in finetuning, we randomly generate 100 conformations and cluster them into 10 conformations, as the model input. A similar pipeline is used in the inference of test data. For most baselines, as they aim to generate conformations from scratch, RDKit-generated conformations are not leveraged. We do not check whether any molecules exist in both pretraining data set and test set of molecular generation. As the same input conformation generation method is used in pretraining and finetuning, and the label of the test set is the accurate conformation generated by semi-empirical density functional theory (DFT)[12], we believe there is no data leakage in the test set.

**Pocket property prediction** The hyperparameters we search are listed in Table 5.

- **Fpocket Score and Druggability Score.** Fpocket tool[5] will output 4 scores, Fpocket score, Druggability score, Total SASA, and Hydrophobicity Score. We call these 4 scores Fpocket scores

Table 5: Search space for pocket property prediction

| Hyperparameter | NRDLD | Fpocket Scores |
|---|---|---|
| Learning rate | [5e-5, 1e-4, 3e-4] | 3e-4 |
| Batch size | [1, 2, 4, 8, 16] | 32 |
| Epochs | 40 | 20 |
| Pooler dropout | [0, 0.1, 0.2, 0.3] | 0 |
| Warmup ratio | [0.0, 0.1] | 0.1 |

Table 6: Performance of Fpocket tool on NRDLD

| | Accuracy | Recall | Precision | F1-score |
|---|---|---|---|---|
| Fpocket score | 0.73 | 0.83 | 0.76 | 0.79 |
| Druggability Score | 0.78 | 0.83 | 0.83 | 0.83 |

(an "s" here). Specifically, the Fpocket score is a custom score by Fpocket; the druggability score is an empirical score calculated from evolution and homologous information. Besides, to verify the effectiveness of the Fpocket tool on real world data, we test this tool on NRDLD. Table6 shows the performance of Fpocket tool on NRDLD dataset.

**Protein-ligand binding pose prediction**

- **Data split** The training set is PDBbind General set v.2020 excluding the complexes covered by CASF-2016. We perform data preprocessing, such as adding missing atoms to both proteins and ligands and manually fixing file-loading errors, before constructing the training set. And we additionally filter the complexes based on the number of residues contained in the pockets (>= 5 ), resulting in a training set of 18k protein-ligand complexes. The test set is CASF-2016, which contains 285 protein-ligand complexes.

- **Binding pose model architecture** As shown in Figure 1, the binding pose model is an encoder-decoder architecture consisting of two 15 layers Uni-Mol as encoder and a 4 layers Uni-Mol as decoder. The decoder Uni-Mol block follows the same setting as the pretraining ones.

- **Scoring function** To evaluate the docking power of our proposed Uni-Mol model, we construct a scoring function, composed of cross distance loss and self-distance loss, out of Uni-Mol. Cross distance loss evaluates the atom-wise distance between atoms on the pocket and ligand, and self-distance evaluates the atom-wise distance between atoms on the same ligand. The ultimate scoring function is a weighted sum of the cross distance loss and the self-distance loss, and the weights are 1.0 and 5.75 respectively.

- **Hyperparameter settings**
  As shown in Figure 1, Uni-Mol directly predicts protein-ligand cross distance and self-distance with MSE loss during finetuning. Dist_threshold is used to mask distances, since atoms that are more than a certain distance apart do not have interactions that would affect the binding pose. We use 10 randomly generated molecular conformations as data augmentation when sampling. Also, a lower dist_threshold is used to reduce variance in sampling with consideration of error in prediction. The details of hyperparameters are shown in Table 7.

- **Exhaustiveness search** To ensure that the comparison between Uni-Mol and popular molecular docking software is unbiased, we increase the exhaustiveness of the global search (roughly proportional to time) of the molecular docking software to observe the effect of computational complexity to docking power on CASF-2016 benchmark. And we find that when exhaustiveness is above 16, the popular molecular docking software can no longer improve the performance by increasing the computational complexity.

- **Differential evolution algorithm** We use a differential evolution algorithm inspired by Deep-dock[19] in protein-ligand pairs. We sample 10 RDKit conformations from the uniform dihedral angle in rotatable bonds, then choose the lowest score function in evolution sampling as the final predicted ligand pose. Moreover, we also tried a faster method, by directly back-propagation from distance-based scoring function to input coordinates.

Table 7: Hyperparameters setup for binding pose prediction

| Hyperparameters for finetuning | Value |
|---|---|
| Learning rate | 3e-4 |
| Batch size | 32 |
| Epochs | 50 |
| Warmup ratio | 0.06 |
| Dropout | 0.2 |
| Dist_threshold | 8.0 |
| Cross distance loss function and weight | MSE, 1.0 |
| Holo distance loss function and weight | MSE, 1.0 |

| Hyperparameters for sampling | Value |
|---|---|
| Population size | 150 |
| Max iterations | 500 |
| Dist_threshold | 5.0 |
| Mutation | (0.5, 1.0) |
| Recombination | 0.9 |
| Conformation size | 10 |
| Cross distance weight | 1.0 |
| Holo distance weight | 5.75 |

Table 8: Exhaustiveness study of popular docking tools on CASF-2016

| | | Ligand RMSD | | | |
|---|---|---|---|---|---|
| | | % Below Threshold ↑ | | | |
| Methods | Exhaustiveness | 0.5 Å | 1.0 Å | 1.5 Å | 2.0 Å |
| Autodock Vina | 1 | 21.40 | 35.79 | 47.02 | 52.28 |
| Autodock Vina | 8 | 23.86 | 44.21 | 57.54 | 64.56 |
| Autodock Vina | 16 | 25.61 | 45.96 | 60.70 | 66.67 |
| Autodock Vina | 32 | 25.96 | 45.96 | 60.00 | 66.32 |
| Vinardo | 1 | 16.84 | 33.33 | 43.16 | 49.82 |
| Vinardo | 8 | 23.51 | 41.75 | 57.54 | 62.81 |
| Vinardo | 16 | 23.51 | 45.26 | 60.70 | 66.67 |
| Vinardo | 32 | 23.86 | 44.56 | 59.30 | 65.61 |
| Smina | 1 | 23.51 | 39.65 | 50.53 | 56.14 |
| Smina | 8 | 23.51 | 47.37 | 59.65 | 65.26 |
| Smina | 16 | **28.77** | 49.47 | 61.40 | 67.72 |
| Smina | 32 | 28.07 | 51.23 | 61.75 | 67.37 |
| Autodock4 | 1 | 4.91 | 18.95 | 26.67 | 28.87 |
| Autodock4 | 8 | 7.02 | 21.75 | 31.58 | 35.44 |
| Autodock4 | 16 | 6.32 | 24.56 | 34.04 | 38.95 |
| Autodock4 | 32 | 6.32 | 23.16 | 34.04 | 38.25 |
| Uni-Mol$_{random}$ | - | 14.04 | 49.47 | 65.26 | 75.44 |
| **Uni-Mol** | - | 24.91 | **70.53** | **84.21** | **88.07** |

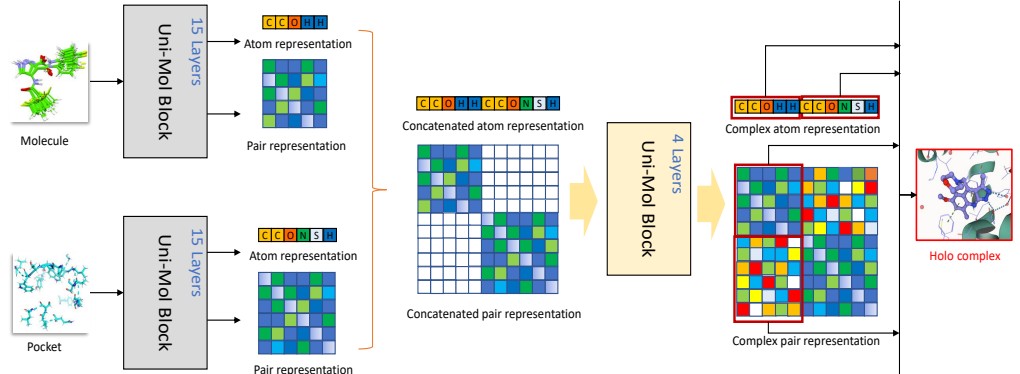

Figure 1: protein-ligand binding pose model: 1) Encoder: molecular representation and pocket representation are obtained from their own pretraining Uni-Mol models; 2) Decoder: representation is concatenated with atom and pair-level, as inputs of a 4 layers Uni-Mol block learning from scratch. 3) Output: The complex representation is used as a project layer to learn the pair distances of molecule and pocket.

**Other details**

- **Max atoms** We use the max atom as 256 because it is enough for the pocket (cover 99.998% pockets ). 256 is not a hard limit. During training, with gradient checkpointing, we can easily extend the atom number to more than 800, by the V100 GPU with 32G memory. There are some recent works that can also significantly reduce the memory cost in Transformer, like Flash-Attention[20]. So we believe the max number of atoms will not be a limit. Besides, with an appropriate sampling strategy, even if the number of atoms could be limited in training time, we can use much more atoms at inference time and still achieve good performance. For example, in Alphafold[21], the training only samples 256/384 residues for saving memories and efficiency, but the inference can use thousands of residues.

- **Vocabulary size.** Vocabulary size is different between molecules and proteins. Because the models for molecules and pockets are different; they don't need to share the same vocabulary. And the vocabulary is made based on the atoms' statistical information in the data. In pocket data, there are amino acids, whose atoms are mostly C, N, O, S and H. While in molecule data, the atom types are more diverse, so a larger vocabulary is used.

## 4 Metrics

In the conformation generation task, following previous work [22, 23], we use the Root of Mean Squared Deviations (RMSD) of heavy atoms to evaluate the difference between the generated conformation and the reference one. Before computing RMSD, the generated conformation is first aligned with the reference one, and the function $\Phi$ aligns conformations by applying rotations and translations to them:

$$\text{RMSD}(\boldsymbol{R}, \hat{\boldsymbol{R}}) = \min_{\Phi}(\frac{1}{n}\sum_{i=1}^{n}||\Phi(\boldsymbol{R}_i) - \hat{\boldsymbol{R}}_i||^2)^{\frac{1}{2}} \tag{1}$$

where $\boldsymbol{R}$ and $\hat{\boldsymbol{R}}$ are the generated and reference conformation, $i$ is the $i$-th heavy atom, and $n$ is the number of heavy atoms.

We use Coverage (COV) and Matching (MAT) to evaluate the performance of the conformation generation model. Higher COV means better diversity, while lower MAT means higher accuracy. Formally, COV and MAT are denoted as:

$$\text{COV}(S_g, S_r) = \frac{\left|\left\{\boldsymbol{R} \in S_r | \text{RMSD}(\boldsymbol{R}, \hat{\boldsymbol{R}}) < \delta, \hat{\boldsymbol{R}} \in S_g\right\}\right|}{|S_r|} \tag{2}$$

Table 9: Ablation studies, molecular property prediction classification tasks

| | Classification (ROC-AUC %, higher is better ↑) | | | | | | | | |
|---|---|---|---|---|---|---|---|---|---|
| Datasets | BBBP | BACE | ClinTox | Tox21 | ToxCast | SIDER | HIV | PCBA | MUV |
| Uni-Mol w/o pair-type | 66.3(1.7) | 76.2(0.2) | 87.1(2.3) | 72.4(0.1) | 62.3(0.4) | 61.2(1.1) | 75.8(0.5) | 85.1(0.1) | 80.9(0.6) |
| Uni-Mol w/o pretraining | 69.0(0.7) | 80.9(5.4) | 68.3(2.2) | 75.8(0.4) | 63.8(0.1) | 61.9(0.5) | 76.2(2.4) | 86.1(0.5) | 62.8(4.0) |
| Uni-Mol w/o pair representation | 71.6(1.3) | 85.4(2.7) | 85.5(1.7) | 79.4(0.1) | 69.3(0.1) | 64.3(0.9) | 80.2(0.2) | 88.4(0.1) | 71.0(7.7) |
| 2D shortest path encoding | 71.6(2.1) | 85.6(1.1) | 83.6(4.0) | 79.6(0.7) | 68.8(0.8) | 63.7(0.1) | 78.9(0.4) | 88.0(0.2) | 78.2(0.6) |
| 1D relative positional encoding | 70.3(1.9) | 77.8(3.7) | 64.2(2.0) | 73.3(0.7) | 64.9(0.2) | 61.5(1.6) | 75.6(0.3) | 77.2(1.4) | 68.7(1.0) |
| Point Transformer | 72.0(0.6) | 84.1(1.3) | 66.9(2.2) | 79.1(0.6) | 65.3(0.3) | 64.3(0.6) | 79.2(0.5) | 87.2(0.4) | 78.1(0.9) |
| Uni-Mol | **72.9(0.6)** | **85.7(0.2)** | **91.6(0.6)** | **79.6(0.5)** | **69.6(0.1)** | **65.5(1.0)** | **80.8(0.3)** | **88.5(0.1)** | **82.1(1.3)** |

Table 10: Ablation studies, molecular property prediction regression tasks

| | Regression (lower is better) | | | | | |
|---|---|---|---|---|---|---|
| | RMSE | | | MAE | | |
| Datasets | ESOL | FreeSolv | Lipo | QM7 | QM8 | QM9 |
| Uni-Mol w/o pair-type | 0.977(0.007) | 2.053(0.053) | 0.951(0.056) | 45.9(1.7) | 0.0156(0.0001) | 0.00473(0.00004) |
| Uni-Mol w/o pretraining | 0.924(0.037) | 1.880(0.206) | 0.745(0.012) | 45.2(0.6) | 0.0174(0.0002) | 0.00653(0.00040) |
| Uni-Mol w/o pair representation | 0.807(0.027) | 1.681(0.068) | 0.611(0.004) | 45.2(1.0) | 0.0158(0.0001) | 0.00573(0.00004) |
| 2D shortest path encoding | 0.831(0.007) | 1.694(0.070) | 0.605(0.003) | 60.6(0.2) | 0.0164(0.0001) | 0.00650(0.00001) |
| 1D relative positional encoding | 0.929(0.035) | 2.237(0.074) | 0.866(0.004) | 77.5(2.7) | 0.0283(0.0007) | 0.02283(0.00078) |
| Point Transformer | 0.828(0.011) | 1.672(0.061) | 0.668(0.007) | 47.2(0.7) | 0.0208(0.0002) | 0.00913(0.00009) |
| Uni-Mol | **0.788(0.029)** | **1.620(0.035)** | **0.603(0.010)** | **41.8(0.2)** | **0.0156(0.0001)** | **0.00467(0.00004)** |

$$\mathrm{MAT}(S_g, S_r) = \frac{1}{|S_r|} \sum_{\boldsymbol{R} \in S_r} \min_{\hat{\boldsymbol{R}} \in S_g} \mathrm{RMSD}(\boldsymbol{R}, \hat{\boldsymbol{R}}) \tag{3}$$

where $S_g$ and $S_r$ are the set of generated and reference conformations, respectively, and $\delta$ is a given RMSD threshold. Following previous work [10, 11], for GEOM-QM9, the threshold is 0.5Å, and for GEOM-Drugs, the threshold value is 1.25Å.

## 5 Ablation studies

### 5.1 Pair-type aware affine module

We investigate the impact of the pair-type aware affine (PTAA) module on the molecular property prediction tasks. As described in Sec 2.1, in invariant spatial positional encoding, the PTAA is combined with the pair Euclidean distance matrix. Tables 9 and 10 show the results of the ablation studies, and we can find that PTAA largely improves the performance of molecular property prediction. There are several possible reasons: 1) in chemicals (and physics), the interactions between two atoms are determined by their distances and types together. Given pair distance and their types, the model can distinguish different interactions, such as Van der Waals forces, covalent interactions, etc., and thus perform better. 2) PTAA enlarges the capacity of pair representation by introducing more trainable parameters, and therefore, the model learns better pair interactions in 3D space and thus performs better.

### 5.2 Pretraining, pair representation and invariant spatial positional encoding

We investigate the impact of pretraining, pair representation and invariant spatial positional encoding on the molecular property prediction tasks. Specifically, to demonstrate the effectiveness of introducing 3D information, we replace the original invariant spatial position encoding with a 2D Graphormer-like[24] shortest path positional encoding and a 1D BERT-like[25] relative position encoding on atoms. For other 3D Transformer baseline, we design an experiment for comparison. Specifically, we replace the spatial encoding method used in Uni-Mol with the one used in Point Transformer[26]. The results are summarized in the following table. Tables 9 and 10 show the results of the ablation studies, and we can find that pretraining, pair representation and invariant spatial positional encoding all largely improves the performance of molecular property prediction. It is clear that 3D information indeed helps the performance of downstream tasks. And compared with Point Transformer, Uni-Mol performs better.

## 6 Training Stability

With Pre-LayerNorm [27] backbone and mixed-precision training, the pretraining sometimes diverges. After investigation, we found there are large numerical values in the intermediate states when divergence happens. We hypothesize that the Final-LayerNorm layer in the Pre-LayerNorm backbone results in the problem. Specifically, Final-LayerNorm is applied to the sum of all encoder layers, denoted as

$$o_i = \text{LayerNorm}(s_i), \quad s_i = \sum_{l=1}^{L} o_i^l \tag{4}$$

where $L$ is the number of layers, $o_i^l$ is the output of the $i$-th position in the $l$-th layer, and $o_i$ is the final output of the $i$-th position, after Final-LayerNorm. Therefore, due to normalization, $s_i$ can be arbitrarily large (or arbitrarily small), without affecting model results. However, a too large or too small numerical value will cause the numerical unstable, especially in the mixed-precision training. To tackle this, we introduce a simple loss, to restrict the value range of $s_i$. Formally, the loss is denoted as

$$\mathcal{L}_{norm} = \text{mean}_i \left( \max \left( \left| \|s_i\| - \sqrt{d} \right| - \tau, 0 \right) \right), \tag{5}$$

where $d$ is the dimension size of $s_i$, $\tau$ is the tolerance factor. In Uni-Mol, we set $\tau = 1$, and both atom-level and pair-level representations are constrained by this loss. Besides, to avoid affecting other loss functions, we set a very small loss weight (0.01) to $\mathcal{L}_{norm}$.

## 7 Related work

**Pretraining**  In recent years, pretraining [28, 29, 30] has received much attention and has been prevailing in many applications. The masked language models, for example, BERT [25] and GPT [31, 32, 33], mask part of the input and predict the masked part to train the model, which has achieved good performance in Natural Language Processing (NLP). There are also works in Computer Vision (CV) inspired by the success of pretraining Transformer in NLP, such as ViT [34] and BEiT [35], applying masking strategy to images to help model training. Recently, some works [36, 37] focus on self-supervised learning that uses the data augmentation strategy to improve the model performance.

**Protein representation learning**  Protein representation learning is critical for drug design. In recent years, many pretraining based methods have been proposed [38, 39, 40]. Besides, the structure of a protein influences how it behaves when bound to a drug-like molecule. Some works also focus on learning from protein 3D structure [21, 41, 42] expecting better performance in 3D structure-related downstream tasks such as protein-ligand binding pose prediction.

**Comparison with Equibind**  For the protein-ligand binding pose prediction task, there are several graph deep learning based methods like Equibind [43]. However, we cannot have an apple-to-apple comparison with Equibind, due to Equibind being proposed for Blind Docking. While Uni-Mol is currently designed for Targeted Docking, which follows most previous traditional tools in docking [44]. The difference is that Blind Docking uses whole protein for docking, while Target Docking directly uses the pocket. We will extend Uni-Mol to Blind Docking tasks in future work.

## 8 Self-attention map visualization

For better interpretability, we conduct a visualization on the self-attention map and pair distance of the molecule as shown in Figure 2. Figure 2 shows that when two atoms in a molecule are close, i.e., the distance between them is small, their corresponding attention weight is large.

## 9 Motivation for using Transformer

Transformer is widely used as a backbone model in representation learning. In recent years, Transformer has shown its power in graph data. For example, Graphormer[24] won two champions at KDD CUP 2021 graph level track and NeurIPS 2021 Open Catalyst Challenge. And some previous works also use Transformer in molecular representation learning, like GROVER[45]. One more

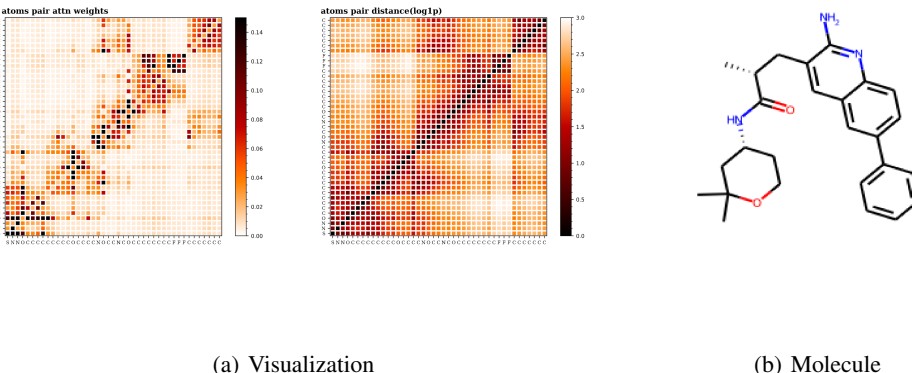

(a) Visualization                    (b) Molecule

Figure 2: Visualization on self-attention map and pair distance of a molecule

motivation is that Transformer has a larger receptive field, as the nodes/atoms are fully connected. While in graph neural networks (GNNs), we usually cut off the edges by locality (distances, bonds). We believe the larger receptive field has more advantages in self-supervised pertaining, as it could learn the long-range interactions from large-scale unlabeled data. For example, in the last row of the attention visualization in Figure 2, there are some columns (21-27) that have slightly large attention weights, while the distances are also large.

**Comparison with Graphormer**    Graphormer motivated us to use Transformer, and we also follow its simplicity in designing the Uni-Mol backbone model. However, the positional encoding (shortest path) used in Grahpormer can only handle 2D molecular graphs, not 3D positions. So we added several modifications to make the model have the ability to handle 3D inputs and outputs. Further, there is a following-up work called 3D-Graphormer [46], adapting this method to 3D molecules. There are several differences between us: 1) Both Uni-Mol and 3D-Graphormer use the pair-wise Euclidean distance and Gaussian kernel to encode 3D spatial information. However, 3D-Graphormer has an additional node-level centrality encoding, which is the sum of spatial encodings of each node. 2) 3D-Graphormer doesn't have pair-representation. 3) Our SE(3) Coordinate Head is different from the "node-level projection head" in 3D-Graphormer. The method used in 3D-Graphormer is an attention layer for 3 axes in 3D coordinate. 4) 3D-Graphormer is not designed for self-supervised pretraining.