# OpenReview forum: "Uni-Mol: A Universal 3D Molecular Representation Learning Framework"
_NeurIPS.cc/2022/Conference — NeurIPS 2022 Submitted_

### Official Review · Reviewer_Ms4D · 2022-07-06

**Rating:** 4
**Confidence:** 4
**Soundness:** 2 fair
**Presentation:** 2 fair
**Contribution:** 2 fair

**Summary:**

The authors proposed the same SE(3)-equivariant transformer variant, called Uni-Mol, for both molecular pretraining and protein pocket pretraining. To make the model SE(3)-equivariant, they devise an invariant spatial positional encoding for the transformer variant.  Specifically, a affine transformation is used to fuse euclidean distance of atom pairs and types of atom pairs, then a Gaussian density function is used to get continuous representations of the atom pairs. After that, the self-attention mechanism takes the pair representation as a bias term in softmax calculation to better leverage 3D information of atoms. Following the idea of EGNN, the authors also propose a SE(3)-equivariant coordinate head to to directly predict coordinates of atoms. According to different downstream tasks, the two models are used independently or jointly.

**Questions:**

1、For molecular conformation generation task, what is the input of the model for fine-tuning? That is, does the model need some kind of ‘random atomic coordinates’ as input for molecular conformation optimization? Moreover, how to randomly generate 10 conformations as the authors mention in Sec. 3.2? I guess ETKGD is utilized again to generate coordinates as inputs of Uni-Mol. If so, do other models need 3d coordinates as input, i.e., using ETKGD in a similar way? Do the authors check whether any molecules exist in both pretraining data set and test set?
2、For pocket property prediction task, the authors construct a regression dataset based on Fpocket tool. What is the difference between Fpocket Score and Druggability Score? Moreover, it seems only Fpocket Score is conducted in Table 4. Besides, how well Fpocket tool can perform in reality? Can the authors test this tool on NRDLD ( if possible )?
3、For 3D prediction tasks of protein-ligand pairs, the authors claim that they use a simple differential evolution algorithm to sample and optimize the complex conformations. More details should be revealed.
4、The authors should investigate the impact of the pair Euclidean distance matrix in ablation studies.
5、The authors should demonstrate the effectiveness of pretraining process. That is, how much improvement can pretraining bring to model performance?
6、The authors mention that 3D spatial positional encoding leaks chemicl bonds in Sec. 2.2 which is contradictory to some content in Sec. 2.1 (third to last sentence in invariant spatial positional encoding part).
7、In Table 2 in supplementary material, max number of atoms is 256 for protein pocket. Is this value big enough for a protein? Moreover, why vocabulary size is different between molecules and proteins (30 for molecule and 9 for pocket )?



**Limitations:**

1、 The proposed SE(3)-equivariant transformer architecture has a limitation on the length of input molecules. In another word, it can not handle with molecules or protein pockets with length bigger than the max atom numbers (in this paper, the number is 256).

**Strengths And Weaknesses:**

This manuscript describes a universal pretrained model for both molecular and protein pocket representation. Specifically, it shows an interesting way of integrating attention mechanism with pair Euclidean distance matrix and atom pair type matrix, keeping the transformer variant SE(3)-equivariant. The authors conduct many downstream tasks to evaluate the performance of the pretrained model and the results are mostly promising.  However, it is a little pity that some aspects of the approach are not clearly communicated.

---

> ### Author Response · Authors · 2022-08-02
> **Response for Reviewer Ms4D**
>
> Thank you very much for the careful review! We will revise our paper to address your comments.
>
> **Max atoms.**  Please note that we use the max atom as 256 because it is enough for the pocket (cover 99.998% pockets ). 256 is not a hard limit. During training, with gradient checkpointing, we can easily extend the atom number to 800+, by the V100 GPU with 32G memory. There are some recent works that can also significantly reduce the memory cost in Transformer, like Flash-Attention[1]. So we believe the max number of atoms will not be a limit. Besides, with an appropriate sampling strategy, even if the number of atoms could be limited in training time, we can use much more atoms at inference time and still achieve good performance. For example, in Alphafold[2], the training only samples 256/384 residues for saving memories and efficiency, but the inference can use thousands of residues.
>
> **For molecular conformation generation task's data.** Thank you for pointing this out, and we will add more details in the paper about this. Yes, we leverage RDKit (ETKGD) for generating inputs in molecular conformation generation tasks. Specifically, in finetuning, we randomly generate 100 conformations and cluster them into 10 conformations, as the model input. A similar pipeline is used in the inference of test data. For most baselines, as they aim to generate  conformations from scratch,  RDKit-generated conformations are not leveraged.  We did not check whether any molecules exist in both pretraining data set and test set of molecular generation. As the same input conformation generation method is used in pretraining and finetuning, and the label of the test set is the accurate conformation generated by semi-empirical density functional theory (DFT)[3], we believe there is no data leakage in the test set.
>
> **Fpocket Score and Druggability Score.** Fpocket tool [4] will output 4 scores, Fpocket score, Druggability score, Total SASA, and Hydrophobicity Score. We call these 4 scores  Fpocket scores (an "s" here). Specifically, the Fpocket score is a custom score by Fpocket; the druggability score is an empirical score calculated from evolution and homologous information.
>
> Performance of Fpocket tool on NRDLD dataset.
>
> | |Accuracy|Recall|Precision|F1-score|
> :---:|:--:|:---:|:---:|:--:|
> Fpocket score|0.73 |0.83|0.76|0.79|
> Druggability Score|0.78|0.83|0.83|0.83|
>
> **Differential evolution algorithm used in protein-ligand pairs.** We use a differential evolution algorithm inspired by Deepdock[5]. We sample 10 RDKit conformations from the uniform dihedral angle in rotatable bonds, then choose the lowest score function in evolution sampling as the final predicted ligand pose. Moreover, we also tried a faster method, by directly back-propagation  from distance-based scoring function to input coordinates. Sorry for missing the details. We will add them in the next version of the paper.

---

> > ### Author Response · Authors · 2022-08-02
> > **Response for Reviewer Ms4D [cont'd]**
> >
> > **Ablation studies for pretraining and pair distance.** Sorry for missing the ablation studies. We have conducted ablation experiments for pretraining and positional encoding. To demonstrate the effectiveness of pair distance, we replace the original invariant spatial position encoding with a 2D Graphormer-like[6] shortest path positional encoding and a 1D BERT-like[7] relative position encoding on atoms. To demonstrate the effectiveness of pretraining, we train our model from scratch on the downstream dataset. The results are summarized in the following table.
> >
> > Dataset|BBBP(%, ↑)|BACE(%, ↑)|QM7(↓)|QM8(↓)|
> > :---:|:--:|:---:|:---:|:--:|
> > **Uni-Mol**|**72.9(0.6)** |**85.7(0.2)**|**41.8(0.2)** |**0.0156(0.0001)**|
> > w/o pretrain|69.0(0.7) |80.9(5.4)|45.2(0.6)|0.0174(0.0002)|
> > 2D shortest path encoding (Graphormer like)|71.6(2.1)|85.6(1.1)|60.6(0.2)|0.0164(0.0001)|
> > 1D BERT-like relative positional encodings on atoms|70.3(1.9)|77.8(3.7)|77.5(2.7)|0.0283(0.0007)|
> >
> > From the above table, it is clear that pretraining and 3D information indeed helps the performance of downstream tasks. Due to the tight timeline, we cannot finish all downstream tasks. We will add the full results to the paper in the next revision.
> >
> > **Contradictory contents about chemical bonds.** Thank you for pointing this out, we will make it clearer in the next version. They are actually not contradictory contents. First, a chemical bond is composed of two atoms and a bond. And a bond has its type and length. For example, C-H is a single bond between a C atom and an H atom, and its bond length is about 1.09 Å. Therefore, although chemical bonds are not directly saved in the 3D spatial encoding, we can easily infer the chemical bond based on the pair type and pair distance. For example, if the distance between a C atom and an H atom is 1.09  Å, we can easily infer that there is a C-H bond. And when one atom is masked, it is also very easy to infer its atom type based on the pair distance.
> >
> > **Vocabulary size.**  First, the models for molecules and pockets are different; they don't need to share the same vocabulary. And the vocabulary is made based on the atoms' statistical information in the data. In pocket data, there are amino acids, whose atoms are mostly C, N, O, S and H. While in molecule data, the atom types are more diverse, so a larger vocabulary is used.
> >
> > Thank you very much for reading. Please reconsider your rating on this paper if your questions and concerns are addressed.
> >
> > >Reference：
> >
> > [1] Dao T, Fu D Y, Ermon S, et al. FlashAttention: Fast and Memory-Efficient Exact Attention with IO-Awareness
> >
> > [2] Jumper J, Evans R, Pritzel A, et al. Highly accurate protein structure prediction with AlphaFold
> >
> > [3] Axelrod S, Gomez-Bombarelli R. GEOM, energy-annotated molecular conformations for property prediction and molecular generation
> >
> > [4] Fpocket: An open source platform for ligand pocket detection
> >
> > [5] A Geometric Deep Learning Approach to Predict Binding Conformations of Bioactive Molecules
> >
> > [6] Ying, Chengxuan, et al. "Do transformers really perform badly for graph representation?."
> >
> > [7] Devlin J, Chang M W, Lee K, et al. Bert: Pre-training of deep bidirectional transformers for language understanding

---

### Official Review · Reviewer_hrSe · 2022-07-11

**Rating:** 6
**Confidence:** 5
**Soundness:** 4 excellent
**Presentation:** 2 fair
**Contribution:** 4 excellent

**Summary:**

This work provides representation learning for 3D molecules (like BERT in NLP). The authors provide two pretrained models using the same model architecture. The first one is trained on 209M molecular conformations, and the second one is trained on 3M protein pocket data. The model architecture is a modified Transformer, and the pretraining tasks include masked atom prediction and 3D position denoising. The pretrained models can be used for various tasks, such as molecular property prediction, molecular conformation generation, protein-ligand binding pose prediction, and pocket druggability prediction. The performance is very good.

**Questions:**

1. What is the difference between the proposed model architecture and existing Transformer architectures like Graphormer?
2. I suggest the authors compare (or discuss) the newly-proposed graph DL-based methods on the protein-ligand binding pose prediction task.
3. Can we extend protein pockets to whole proteins since in some cases, we don’t know the pocket? If not, what is the challenge?

**Limitations:**

The authors have addressed the limitations of their work.

**Strengths And Weaknesses:**

Strengths:
1. This work is well-written and easy to follow.
2. It provides two good pretrained models for small molecules and protein pockets. The models can be finetuned for various downstream tasks.
3. The experiment results show that the provided models can outperform previous methods on various tasks.

Weakness:
1. About the model architecture: There are many existing Transformer architectures like [1] for graph representations. The authors should discuss the differences with these methods.
[1] Ying, Chengxuan, et al. "Do transformers really perform badly for graph representation?."
2. About baseline results: For the protein-ligand binding pose prediction task, there are several graph DL-based methods like [2], I suggest the authors include such papers.
[2] Stärk, Hannes, et al. "Equibind: Geometric deep learning for drug binding structure prediction."

---

> ### Author Response · Authors · 2022-08-02
> **Response for Reviewer hrSe**
>
> Thank you very much for supporting our work and careful review! We will revise our paper to address your comments.
>
> **Comparison with Graphormer.** Graphormer[1] motivated us to use Transformer, and we also follow its simplicity in designing the Uni-Mol backbone model. However,  the positional encoding (shortest path) used in Grahpormer can only handle 2D molecular graphs, not 3D positions. So we added several modifications to make the model have the ability to handle 3D inputs and outputs. Very sorry, we found the citation to Graphormer in the paper was accidentally removed in paper revisions, and we will add it back in the next version.
>
> **Comparison with newly-proposed graph DL-base methods on the protein-ligand binding pose prediction task.** Thank you for the suggestion. Equibind[2] is an excellent work, and we will add a discussion about it in the paper. However, we cannot have an apple-to-apple comparison, due to Equibind being proposed for Blind Docking. While Uni-Mol  is currently designed for Targeted Docking, which follows most previous traditional tools in docking[3]. The difference is that Blind Docking uses whole protein for docking, while Target Docking directly uses the pocket.  We will extend Uni-Mol to Blind Docking tasks in  future work.
>
> **Extend to the whole protein docking.**  This is a very good question. We can follow the two-stage solutions in the traditional docking pipeline: first detect a pocket, by tools like Fpocket[4], or by human, then perform the Targeted Docking in the specific pocket. We can also extend the Uni-Mol framework to support Blind Docking like Equibind. To do that, we may need to extend the pocket pretraining to the whole protein pretraining, and use the whole protein in the finetuning as well. The challenge is that the number of total atoms in the whole protein could be very large. So we may have to use $C_\alpha$ atoms only.
>
> Thank you very much for reading. Please reconsider your rating on this paper if your questions and concerns are addressed.
>
> >Reference:
>
> [1] Ying, Chengxuan, et al. "Do transformers really perform badly for graph representation?."
>
> [2] Stärk, Hannes, et al. "Equibind: Geometric deep learning for drug binding structure prediction."
>
> [3] GPU-Accelerated Drug Discovery with Docking on the Summit Supercomputer: Porting, Optimization, and Application to COVID-19 Research
>
> [4] Fpocket: An open source platform for ligand pocket detection

---

> > ### Comment · Reviewer_hrSe · 2022-08-06
> > **Reply to authors**
> >
> > Thanks for the authors’ response. I don’t have further questions. Another paper I want to mention is [1], a following-up paper by Graphormer, adapting this method to 3D molecules.
> >
> > Overall, I think this is a good paper and can contribute to the research area of 3D molecules.
> >
> > [1] Shi, Yu, et al. "Benchmarking graphormer on large-scale molecular modeling datasets." arXiv preprint arXiv:2203.04810 (2022).

---

> > > ### Author Response · Authors · 2022-08-07
> > > **Thank you for the response**
> > >
> > > Thank you, we just check the model in your mentioned paper, and the following are the differenes:
> > > 1.  Both Uni-Mol and 3D-Graphormer use the pair-wise Euclidean distance and Gaussian kernel to encode 3D spatial information. However, 3D-Graphormer has an additional node-level centrality encoding, which is the sum of spatial encodings of each node.
> > > 2. 3D-Graphormer doesn't have pair-representation.
> > > 3. Our SE(3) Coordinate Head is different from the "node-level projection head" in 3D-Graphormer. The method used in 3D-Graphormer is an attention layer for 3 axes in 3D coordinate.
> > > 4. 3D-Graphormer is not designed for self-supervised pretraining.
> > >
> > > We will also add the above comparison to the paper. And thank you again for supporting our work!

---

### Official Review · Reviewer_Ue6N · 2022-07-11

**Rating:** 4
**Confidence:** 4
**Soundness:** 2 fair
**Presentation:** 2 fair
**Contribution:** 1 poor

**Summary:**

The paper proposes a pretraining / self-supervised learning method for 3D molecular representation learning. The backbone is Transformer based. The pretraining is adding noise to atom coordinates. The pretraining  Experiments on molecular property prediction, molecular conformation generation, pocket property prediction and protein-ligand binding pose prediction show the effectiveness of the proposed method.


**Questions:**

1. The novelty of the proposed method seems not that significant.
- One difference or advantage that the paper claims is 3D encoding. However, the Euclidean distance map (e.g.,[1]) or Gaussian functions for distance processing [2,3]  has been widely used in protein 3D structure learning.

[1] Highly accurate protein structure prediction with AlphaFold

[2] GraphQA: protein model quality assessment using graph convolutional networks

[3] Learning from Protein Structure with Geometric Vector Perceptrons

- There have been a few methods for the atom-level molecule or protein 3D structure modeling (e.g., [4]). The proposed method should be compared with them via experiments.

[4] Intrinsic-Extrinsic Convolution and Pooling for Learning on 3D Protein Structures

- The pretraining method seems very simple. It is more like a trick.
- (Minor) There are a few existing pretraining / self-supervised learning methods for protein representation learning. The authors might want to add a discussion about them.

[5] Structure-aware Protein Self-supervised Learning

[6] Contrastive Representation Learning for 3D Protein Structures

[7] Protein Structure Representation Learning by Geometric Pretraining


2. The motivation to employ Transformer is not clear. The explanation that "Transformer is the default backbone in representation learning" is not convincing or even incorrect. Moreover, as a Transformer-related work, it could be better to visualize the self-attention map to verify the motivation for employing Transformers.


3. Besides adding the pair representation, is there any other difference with the original Transformer?  Moreover, using Transformers for 3D modeling has been explored in 3D vision. Some important competitors should be compared, such as [8], to show the effectiveness of the proposed method.

[8] Point Transformer.



4. Some important experiments or ablation studies are missing.

a) Accuracy w/ and w/o pertaining.

b) Effect of invariant spatial positional encoding.

c) Effect of pair representation.

d) Effect of pair representation.

e) Effect of SE(3)-equivariance coordinate head.


**Ethics Review Area:**

["I don’t know"]

**Strengths And Weaknesses:**

1. Among the early efforts, the paper proposes a pretraining / self-supervised learning method for universal 3D molecular representation
learning framework.

2. The proposed method achieves SOTAs on several tasks and datasets.

---

> ### Author Response · Authors · 2022-08-02
> **Response for Reviewer Ue6N**
>
> Thank you very much for the careful review! We will revise our paper to address your comments.
>
> **Originality/Novelty of Uni-Mol.**  Please refer to our response to all reviewers ("On the novelty of Uni-Mol").
>
> **Euclidean distance map.** Also, refer to our response to all reviewers. The objective of Uni-Mol is not to develop a backbone model, nor to propose a better 3D encoding method, but to build up a framework for tasks in drug design with a focus on organic molecule drugs, especially in the 3D tasks. Besides, in our paper (L73), as we said "we simply use Euclidean distances of all atom pairs", we did not claim that our advantage/difference is the Euclidean distance map. What we claimed is that Uni-Mol can "directly take 3D positions as both inputs and outputs" (L42), not a new 3D encoding.
>
> **Atom-level baseline models.**    As aforementioned, our objective is not to develop a pretrained backbone model. Besides, most previous atom-level models are designed for supervised learning, not for self-supervised/pretraining, and most of them cannot output the per-atom 3D positions. Therefore, most of the previous work cannot be directly used as the backbone model in Uni-Mol. For example, the paper you posted, "Intrinsic-Extrinsic Convolution and Pooling for Learning on 3D Protein Structures", is a ResNet-based model for classification, which cannot be used in Uni-Mol's downstream tasks which need 3D outputs, like molecular conformation generation and protein-ligand binding pose prediction. We will also add a discussion about this in the paper.
>
> **The pretraining method is very simple.** As mentioned in "On the novelty of Uni-Mol", simplicity does not imply the lack of novelty. Finding a simple and effective solution is usually not easy.  We  tried several pretraining tasks, for example, contrastive learning over different conformations, learning the mapping from 2D<->3D, etc. And we found the simplest masked atom prediction combined with the 3D position denoising task performs very well.
>
> **Discussion with protein representation learning works.** Thank you. We will discuss them in the paper. However, please note that our paper primarily deals with organic molecules, not proteins.
>
> **Motivation to employ Transformer.** Thank you for the suggestion, "Transformer is the default backbone in representation learning" indeed is not correct. We will change it to "Transformer is widely used as a backbone model in representation learning".  And we will add more details about the motivation for using Transformer in the paper. In short, Transformer has shown its power in graph data recently, for example, Graphormer [1] won two champions at KDD CUP 2021 graph level track and NeurIPS 2021 Open Catalyst Challenge. And some previous works also use Transformer in molecular representation learning, like GROVER [2].
>
> **Self-attention map visualization.** Thank you very much for the suggestion, we would like to add the self-attention map visualization to the paper.
>
> **Changes compared with vanilla Transformer.**  Our design principle for the backbone model is as simple as possible, so we don't make many changes. All our modifications are summarised in Sec2.1 of the paper.

---

> > ### Author Response · Authors · 2022-08-02
> > **Response for Reviewer Ue6N [cont'd]**
> >
> > **Point Transformer.**  Similar to previous atom-level models, the transformer models used in 3D vision are mostly designed for supervised learning, not self-supervised learning, and cannot output 3D positions. Therefore, it is hard to compare them directly. However, we still design an experiment for comparison per your request. Specifically, we replace the spatial encoding method used in Uni-Mol with the one used in Point Transformer. The results are summarized in the following table, and it clearly shows that Uni-Mol is better.
> >
> > Dataset|BBBP(%, ↑)|BACE(%, ↑)|QM7(↓)|QM8(↓)|
> > :---:|:--:|:---:|:---:|:--:|
> > **Uni-Mol**|**72.9(0.6)** |**85.7(0.2)**|**41.8(0.2)** |**0.0156(0.0001)**|
> > Point Transformer|72.0(0.6)|84.1(1.3)|47.2(0.7)|0.0208(0.0002)|
> >
> > **Ablation studies.** Sorry for missing the ablation studies and thanks for your suggestion. We have conducted the ablation studies per your request. To demonstrate the effectiveness of invariant spatial positional encoding, we replace it with a 2D Graphormer-like shortest path positional encoding and a 1D BERT-like[3] relative position encoding on atoms. To demonstrate the effectiveness of pretraining, we train our model from scratch on the downstream dataset. And we only reserve the invariant spatial positional encoding and remove the update of pair representation to prove the effectiveness of the pair representation. The results are summarized in the following table.
> >
> > Dataset|BBBP(%, ↑)|BACE(%, ↑)|QM7(↓)|QM8(↓)|
> > :---:|:--:|:---:|:---:|:--:|
> > **Uni-Mol**|**72.9(0.6)** |**85.7(0.2)**|**41.8(0.2)** |**0.0156(0.0001)**|
> > w/o pretrain|69.0(0.7) |80.9(5.4)|45.2(0.6)|0.0174(0.0002)|
> > w/o pair repr|71.6(1.3)|85.4(2.7)|45.2(1.0) |0.0158(0.0001)|
> > 2D shortest path encoding (Graphormer like)|71.6(2.1)|85.6(1.1)|60.6(0.2)|0.0164(0.0001)|
> > 1D BERT-like relative positional encodings on atoms|70.3(1.9)|77.8(3.7)|77.5(2.7)|0.0283(0.0007)|
> >
> > From the above table, it is clear that pretraining, pair representation, and 3D information indeed help the performance of downstream tasks. Due to the tight timeline, we cannot finish all downstream tasks. We will  add the full results to the paper in the next revision.
> >
> > Please note we did not conduct the ablation study for SE(3)-equivariance coordinate head. The SE(3)-equivariance coordinate head is introduced for the ability to output coordinates directly, thus broadening the application scopes of our method. It is not introduced to enhance the effectiveness of the model. Therefore, we think it is not necessary to perform ablation experiments on it.
> >
> >
> > Thank you very much for reading. Please reconsider your rating on this paper if your questions and concerns are addressed.
> >
> > >Reference:
> >
> > [1] Ying, Chengxuan, et al. "Do transformers really perform badly for graph representation?."
> >
> > [2] Rong Y, Bian Y, Xu T, et al. Self-supervised graph transformer on large-scale molecular data
> >
> > [3] Devlin J, Chang M W, Lee K, et al. Bert:
> > Pre-training of deep bidirectional transformers for language understanding

---

> > > ### Comment · Reviewer_Ue6N · 2022-08-08
> > > **Response to authors' rebuttal**
> > >
> > > Thank the authors for their rebuttal and their detailed responses to my review!
> > >
> > > A few comments:
> > >
> > > * Based on the authors' responses regarding the novelty, the main contribution is a *framework*, in which some core components have been used in existing works. First, I would suggest the authors provide a summary of novelty in the paper.  Regarding the current paper writing, readers may misunderstand that the paper focuses on component design. The paper uses many figures or formulations to illustrate positional encoding, self-attention, etc. This may mislead readers about the contribution. Second, if the main contribution is a *framework*, that means the paper is mainly about engineering. I recognize the paper tries to resolve an interesting problem. However, an engineering or a *framework* design may be not that significant to machine learning.
> > >
> > > * It seems that the authors did not provide a revision of the paper.
> > >
> > > * The attention map is not provided. The reviewer understands Transformer has been used in existing works. However, just following them without a clear motivation is not that intereesting.  But anyway, this won't be my concern for the paper.
> > >
> > > *  The reviewer understands the rebuttal time is limited. However, the ablation study should be provided in the initial submission.  Otherwise, the paper might have been considered as not completed.
> > >
> > > * Since the main contribution is a *framework*, which should not depend on specific components,  it could be better to try different components. For example, replacing Transformer with GNN, to show the generalization ability. This would increase the contribution of the proposed *framework*.

---

> > > > ### Author Response · Authors · 2022-08-08
> > > > **Thank you for the further comments!**
> > > >
> > > > We thank the reviewer for the additional comments. Please see our feedbacks as follows.
> > > > - About the paper revision.
> > > >   - As we add many discussions according to the reviewer comments, it exceeds the paper length limit. And we are still waiting for the complete ablation studies to be finished. So we didn't update the paper for now. If you need us to update it, we can put them in the Appendix first, and update the supplementary material. (updated: we just update the supplementary material).
> > > > - About ablation study.
> > > >    - We mainly focus on the real-world downstream applications in our experiment section and we don't think the ablation study about the pretraining is a necessary part for the completeness of the paper. So we don't agree that "paper might have been considered as not completed". However, we do agree that the ablation study is important, as it helps us to better understand the proposed backbone model. We thank the reviewer's comments, which helped us to improve our paper with *5* more ablation studies during the paper review.
> > > >    - We have updated the ablation study results on 15/15 tasks in the Appendix.
> > > > - About the highlight of contribution.
> > > >   - Thank you for the suggestion, we will summarize our contributions in the introduction.
> > > >   - Regard to "many figures or formulations to illustrate positional encoding, self-attention, etc. ", we actually only have 2 figures in the paper, one is the whole framework, and another one is the design of the backbone model.
> > > > - About the motivation for using Transformer
> > > >    - We have added the visualization of attention into supplementary material.
> > > >    - One more motivation is that Transformer has a larger receptive field, as the nodes/atoms are fully connected. While in GNN, we usually cut off the edges by locality (distances, bonds). We believe the larger receptive field has more advantages in self-supervised pertaining, as it could learn the long-range interactions from large-scale unlabeled data. For example, in the last row of the attention visualization, there are some columns (21-27) that have slightly large attention weights, while the distances are also large.
> > > > - We don't agree "if the main contribution is a framework, that means the paper is mainly about engineering. I recognize the paper tries to resolve an interesting problem. However, an engineering or a framework design may be not that significant to machine learning.".
> > > >   - First, our paper is submitted for "Machine Learning for Sciences (e.g. biology, physics, health sciences, social sciences) https://nips.cc/Conferences/2022/CallForPapers". Our goal is to use machine learning to improve the applications in drug design, not to improve machine learning itself.
> > > >   - second, our framework is not a simple engineering work, nor a combination of existing works. Simply using existing models and data for pre-training cannot solve the 3D tasks in drug design. For example, most existing GNN models cannot meet the requirements of output the 3D positions directly and be used in self-supervised tasks simultaneously. Even though the backbone model in Uni-Mol is simple, and some designs in it are inspired by previous works, it is still a new model designed for the entire Uni-Mol framework.
> > > > - We don't agree "Since the main contribution is a framework, which should not depend on specific components, it could be better to try different components. For example, replacing Transformer with GNN, to show the generalization ability. This would increase the contribution of the proposed framework."
> > > >    - In Uni-Mol framework, the backbone model needs to be 1) able to be used in self-supervised training; 2) able to encode 3D inputs; 3) able to output 3D positions. To our best knowledge, no previous model meets these requirements. So a simple replacement is not possible.
> > > >    - Maybe we have a different understanding of the "component" in the framework. In Uni-Mol's framework, the whole backbone model is a standalone "component".  Besides the backbone model, as shown in Figure 1 in the paper, the remaining components are two pertaining datasets, the tasks for self-supervised learning, the downstream tasks, and the methods to use the pretraining models to finetune them, especially for 3D downstream tasks. Therefore, we don't treat the detailed layers inside the backbone model as the components of Uni-Mol. Moreover, as the objective of Uni-Mol is not to develop (or find) a new backbone model, we don't think it is necessary for us to enumerate the possible combination of backbone models.
> > > > - We don't agree "in which some core components have been used in existing works."
> > > >     - In Uni-Mol framework, we create the pertaining dataset by ourselves, design the backbone model and self-supervised task for learning 3D representation and 3D outputs, and design the finetune strategies in the various downstream 3D tasks. To our best knowledge, most of these components are new.

---

> > > > > ### Author Response · Authors · 2022-08-08
> > > > > **About our contribution**
> > > > >
> > > > > We want to highlight again, that our contribution is "to build up a new framework for tasks in drug design with a focus on organic molecule drugs, especially in the 3D tasks, which cannot be covered by previous frameworks."
> > > > >
> > > > > We are not to develop a pretrained backbone model, nor a framework of the combination of existing works, but a new framework that can solve the 3D tasks which cannot be covered by previous frameworks.

---

### Official Review · Reviewer_byGu · 2022-07-11

**Rating:** 6
**Confidence:** 4
**Soundness:** 4 excellent
**Presentation:** 4 excellent
**Contribution:** 3 good

**Summary:**

This paper proposes a method to incorporate 3D information into molecular representation learning, Uni-Mol. Particularly, Uni-Mol includes three parts:

1. Two models (a molecular model and a pocket model) built upon the SE(3)-equivariant transformer architecture;
2. Two large-scale datasets (a 209M molecular conformation dataset and a 3M candidate protein pocket dataset) and corresponding pretraining strategies;
3. Finetuning strategies for various downstream tasks.

In the experiments, Uni-Mol outperforms SOTA in most of the tested tasks and demonstrates its ability in few-shot learning settings.

**Questions:**

* In line 114 and line 119, it is mentioned that the 3D conformations and pocket binding positions are obtained by using simulation or optimization toolkits. Will these toolkits introduce bias to the data? If yes, do you have any idea for alleviating such bias?
* In line 130, it states that for the 3D position denoising task, the noise is uniformly sampled from [-1 A, 1 A]. How is the sampling strategy determined? Have you tried other sampling strategies like using a Gaussian distribution?
* In line 204, it is concluded that 3D information helps the model to learn better representations. Are there any ablation study results that can justify this more convincingly?
* Will the datasets be released for academic use?

**Limitations:**

The limitations are not highlighted or summarized in the paper.

To me, my most concern is about the data used to pretrain the model. As mentioned in the Questions section, since the 3D information in the data mostly comes from computational results instead of ground-truth annotations, how do we know such bias in data won't influence the model performance? If bias is inevitable, what benefits can be guaranteed by using such data?

**Strengths And Weaknesses:**

**Strengths**:

* (originality) The proposed method, Uni-Mol, is original and has reasonable designs: (1) For the backbone model architecture, modifications (including invariant spatial positional encoding, pair representation, and a coordinate head) have been made to the standard Transformer to adapt to 3D molecular representation; (2) Two new large-scale datasets as well as the proposed 3D position denoising task for pretraining; (3) Detailed guidances of how to fine-tune Uni-Mol on downstream tasks.
* (quality) The experiments were conducted thoroughly and touched four aspects of molecular representation learning: molecular property prediction, molecular conformation generation, pocket property prediction, and protein-ligand binding pose prediction. For each task, the datasets and baselines are chosen carefully.
* (significance) The proposed method Uni-Mol outperforms previous work in almost every tested task, which is impressive. In some tasks, like molecular conformation generation and pocket property prediction, the improvements are significant.
* (clarity) The paper is presented clearly and easy to follow. Contributions are properly emphasized.

**Weaknesses**:

* (originality) The design of Uni-Mol is largely based on previous work. For the backbone model architecture, it is not new to use Transformers in molecular representation (e.g., GROVER), and most modifications can also be found in previous work (e.g., pairwise Euclidean distance matrix representation, the SE(3)-equivariant head as in EGNN, the masked atom prediction task). This limits the novelty of the proposed method.
* (significance) Some experiment results didn't show significant improvement over previous methods (e.g., molecular property prediction tasks and ESOL). But this is acceptable since tasks like molecular property prediction have already been studied for long, and the baseline models are very competitive.
* The related work focuses more on molecular representation methods, while the literature on pretraining has been ignored.

---

> ### Author Response · Authors · 2022-08-02
> **Response for Reviewer byGu**
>
> Thank you very much for supporting our work and careful review! We will revise our paper to address your comments.
>
> **Originality/Novelty of Uni-Mol.** Please refer to our response to all reviewers ("On the novelty of Uni-Mol").
>
> **Missing pretraining literature.** We will add more literature on pretraining in the next version.
>
> **3D conformation data related.**
> 1. The candidate pocket dataset was collected from the PDB database of protein structures surface experimentally resolved. L117~L122 describes how we detect, extract, clean, and enhance candidate pockets.
> 2. About the bias in molecular conformation data.
>
>    a. The same molecular conformation generation (computation) pipeline is used in both pretraining and finetuning. Therefore, in most downstream tasks, there is no bias between pretraining and downstream tasks.
>
>    b. The space of molecular conformation is actually quite large, and in different environments, the stable (low-energy) conformations are also different. Therefore, it is non-trivial to get ground-truth annotations of molecular conformations, in both experiment and computation. Therefore, we use multiple conformations (10 in our current setting), to alleviate the possible bias, in both pretraining and finetuning.
>
>    c. To obtain a conformational distribution that fully conforms to the laws of physics, long-term molecular dynamics simulations and conformational optimization at the density functional theory(DFT) level are required. However, it is computationally costly. Therefore, it is almost infeasible to generate large-scale pretraining data by that protocol.  And it is also very inefficient to be used in real-world downstream tasks, most of which don't contain conformations and need to generate conformations on-the-fly.
>
>   To summarize, due to the large space of molecular conformation, we choose  an efficient method to generate multiple conformations for a molecule, to cover the conformation space as much as possible. Besides, it is able to generate a large-scale conformation dataset for pretraining, and be efficiently used in downstream tasks.  Hopefully, the above response can address your concerns about the conformation data.
>
> **Noise sampling strategies.** We tried Gaussian distribution and truncated Gaussian distribution for coordinates noises. The experimental results showed slight decreases in performance for the downstream task. Besides, maybe due to the larger range of noise, Gaussian distribution sometimes caused numerical instability in the fp16 mix-precision training. So we use the uniform distribution.
>
> **Ablation studies for 3D information.**   Sorry for missing the ablation studies. We have run an ablation experiment for this and got some results in the BBBP, BACE, QM7 and QM8 downstream tasks. To demonstrate the effectiveness of introducing 3D information, we replace the original invariant spatial position encoding with a 2D Graphormer-like[1] shortest path positional encoding and a 1D BERT-like[2] relative position encoding on atoms. The results are summarized in the following table.
> Dataset|BBBP(%, ↑)|BACE(%, ↑)|QM7(↓)|QM8(↓)|
> :---:|:--:|:---:|:---:|:--:|
> **Uni-Mol**|**72.9(0.6)** |**85.7(0.2)**|**41.8(0.2)** |**0.0156(0.0001)**|
> 2D shortest path encoding (Graphormer like)|71.6(2.1)|85.6(1.1)|60.6(0.2)|0.0164(0.0001)|
> 1D BERT-like relative positional encodings on atoms|70.3(1.9)|77.8(3.7)|77.5(2.7)|0.0283(0.0007)|
>
> From the above table, it is clear that 3D information indeed helps the performance of downstream tasks. Due to the tight timeline, we cannot finish all downstream tasks. We will  add the full results to the paper in the next revision.
>
> **Data release.**  Yes, we will release all data and codes.
>
> Thank you very much for reading. Please reconsider your rating on this paper if your questions and concerns are addressed.
>
> >Reference:
>
> [1] Ying, Chengxuan, et al. "Do transformers really perform badly for graph representation?."
>
> [2] Devlin J, Chang M W, Lee K, et al. Bert: Pre-training of deep bidirectional transformers for language understanding

---

### Author Response · Authors · 2022-08-02
**On the novelty of Uni-Mol**

We thank all reviewers for your comprehensive and insightful reviews, and have replied to your comments respectively.
There are some comments worrying about the novelty of Uni-Mol. It is sometimes tricky to evaluate the novelty of a work since one can hardly avoid being subjective for such an evaluation -- it's also hard for us to claim it on an absolutely objective matter. Yet we hope to discuss this issue and better explain the novelty of the work here.

It is important to realize that the objective of Uni-Mol is *not* to develop a pretrained backbone model, but to build up a new *framework* for tasks in drug design with a focus on organic molecule drugs, especially in the *3D tasks*, which cannot be covered by previous frameworks. Specifically, previous frameworks mostly focused on molecular property prediction tasks, and achieved competitive performance. Uni-Mol cannot only outperform them in these prediction tasks, but also extend the application scopes to many 3D tasks and achieved SOTA performance as well. Some of the components in the backbone model are inspired from some previous work, like AlphaFold, but we don't think this should affect the novelty of Uni-Mol in its application domain. Taking BERT as an example, will you think it is *not* novel due to its backbone model (Transformer) not being originally proposed by itself?

With this consideration in mind, let us try to address concerns like the *simplicity* of the backbone Transformer model. Simplicity doesn't imply the lack of novelty. Finding a simple and effective solution is usually not easy. There are many efforts behind the presented version of the model architecture (We can add a "Failed Attempts" section in the Appendix if needed). For example, to encode 3D spatial information, we tried several "fancy" methods used in previous works, like SchNet, DIMENet, and GemNet. And we found most of them achieve a similar performance as Gaussian. Following the principle of parsimony, we use the simplest Gaussian. Besides, based on Gaussian, we tried several methods to enhance the distance between different atom types, and found the simple affine transformation worked very well. There are several failed attempts. For example, we tried to explicitly encode chirality into the model, due to the current model cannot distinguish the symmetrical molecules. However, it is only slightly better, so we didn't use it. In the pretraining tasks, we also tried many strategies, for example, contrastive learning over different conformations, learning the mapping between 2D and 3D, etc. And we found the simplest masked atom prediction combined with the 3D position denoising task performs very well.  In short, when two methods perform similarly, we always choose the simple one; when a fancy/complicated method is not necessary and doesn't bring much gain, we will not use it.

Hopefully, the above can address your concerns. If you still have any questions/concerns about the novelty, we can discuss them in this thread.

Thanks,

Authors

---

> ### Author Response · Authors · 2022-08-08
> **About the Uni-Mol "framework" and its "components"**
>
> It seems some reviewers misunderstood the Uni-Mol framework and its "components". So we make a clarification here.
>
> In Uni-Mol's framework, the whole backbone model is a standalone "component". Besides the backbone model, as shown in Figure 1 in the paper, the remaining components are two pertaining datasets, the tasks for self-supervised learning, the downstream tasks, and the methods to use the pretraining models to finetune them, especially for 3D downstream tasks. Therefore, we don't treat the detailed layers/components inside the backbone model as the components of Uni-Mol. Moreover, as the objective of Uni-Mol is not to develop (or find) a backbone model, we don't think it is necessary for us to enumerate the possible combination of backbone models.
>
> We want to highlight again, that our contribution is "to build up a new framework for tasks in drug design with a focus on organic molecule drugs, especially in the 3D tasks, which cannot be covered by previous frameworks."
>
> We are not to develop a pretrained backbone model, nor a framework of the combination of existing works, but a new framework that can solve the 3D tasks which cannot be covered by previous frameworks.

---

### Author Response · Authors · 2022-08-08
**Revision of the paper**

We thank all reviewers for your comprehensive and insightful reviews. Based on previous comments, we provide a revision of the paper, including more details on downstream tasks, more discussions with related works, a visualization of the attention map, and 5 ablation studies.

Due to the paper length limit, we put these contents in the appendix temporarily. We will move them to the main body for the camera-ready version when an additional page will be allowed.

Thanks,

Authors

---

### Author Response · Authors · 2022-08-09
**A gentle reminder to the reviewers for the "Author- Reviewer Discussion"**

Dear reviewers,

Thank you so much for the comprehensive and insightful reviews. Based on the review comments, we made a significant effort to finish additional 5 ablation studies from scratch in the tight discussion period. And we have updated the Appendix with these new results, and more discussions according to review comments. We would very much appreciate it if you would consider raising your score in light of our response.

We thank "Reviewer hrSe" so much for your support and the quick response.

And we also thank "Reviewer Ue6N" so much for the further comments, hope our new response to them and the revision in the Appendix can address your concerns.

We hope we can get the responses from "Reviewer byGu" and "Reviewer Ms4D" soon, as the "Author- Reviewer Discussion" will end in several hours. Your further comments and suggestions will be very helpful for us!

Thanks,

Authors

---

### Meta-Review · Area_Chair_qBBq · 2022-08-26

**Recommendation:** Reject
**Confidence:** Certain

**Metareview:**

This paper proposes a new framework for molecular representation learning (MRL) using both 2D and 3D molecular data. This framework is general and applied to various problems (e.g., protein-ligand binding pose prediction and molecular conformation prediction). I believe this paper is potentially quite impactful and able to reshape how research is conducted for MRL research.

However, the contribution of this paper is unclear in its current form.
- The proposed methodology is not very novel and uses a combination of existing methods.
- While the main contribution of this paper is to propose a new framework for MRL, the experiments focus on evaluating a single algorithm (i.e., SE(3)-equivariant model + 3 self-supervised tasks) compared to existing algorithms under different frameworks. In other words, the experiments do not deliver new information since (1) existing works demonstrated how combining 2D & 3D data improves downstream task performance and (2) pretraining is useful for the considered downstream tasks.

Overall, I recommend rejection for this paper. However, I believe this paper can be a very strong submission for the next conference if the authors clearly demonstrate their contribution. For example, I think the proposed idea would be pleasantly presented as an important "benchmark" paper, rather than a framework with superior performance.

**Award:**

No

---

### Decision · Program_Chairs · 2022-09-14

Reject